# RIFLE: REMOVAL OF IMAGE FLICKER-BANDING VIA LATENT DIFFUSION ENHANCEMENT

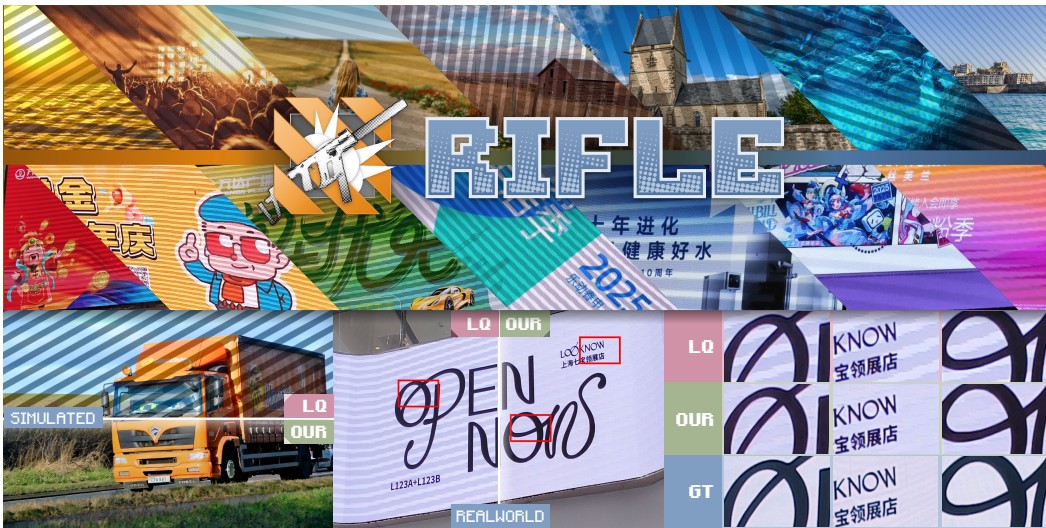

Figure 1: Overview of **RIFLE**. The top part shows the datasets we build, including the simulated training dataset (first row) and the real-world testing dataset (second row). The bottom part shows the flicker-banding removal effect on simulated and real-world images.

## ABSTRACT

Capturing screens is now routine in our everyday lives. But the photographs of emissive displays are often influenced by the *flicker-banding* (FB), which is alternating bright–dark stripes that arise from temporal aliasing between a camera's rolling-shutter readout and the display's brightness modulation. Unlike moiré degradation, which has been extensively studied, the FB remains underexplored despite its frequent and severe impact on readability and perceived quality. We formulate FB removal as a dedicated restoration task and introduce **R**emoval of **I**mage **F**licker-**B**anding via **L**atent Diffusion **E**nhancement, *RIFLE*, a diffusion-based framework designed to remove FB while preserving fine details. We propose the *flicker-banding prior estimator (FPE)* that predicts key banding attributes and injects it into the restoration network. Additionally, *Masked Loss (ML)* is proposed to concentrate supervision on banded regions without sacrificing global fidelity. To overcome data scarcity, we provide a *simulation pipeline* that synthesizes FB in the luminance domain with stochastic jitter in banding angle, banding spacing, and banding width. Feathered boundaries and sensor noise are also applied for a more realistic simulation. For evaluation, we collect a *paired real-world FB dataset* with pixel-aligned banding-free references captured via long exposure. Across quantitative metrics and visual comparisons on our real-world dataset, RIFLE consistently outperforms recent image reconstruction baselines from mild to severe flicker-banding. To the best of our knowledge, it is *the first work* to research the simulation and removal of FB. Our work establishes a great foundation for subsequent research in both the dataset construction and the removal model design. Our dataset and code will be released soon.

## 1 INTRODUCTION

Capturing screens has become routine in everyday life: (i) students photograph lecture slides on projectors, (ii) professionals document dashboards and error messages on monitors, (iii) creators share phone or smartwatch interfaces, (iv) commuters record LED billboards or vehicle clusters, and so on. Despite impressive progress in mobile imaging, photographs of emissive displays remain prone to characteristic degradations. Among these, moiré (Zhang et al. (2023); Xu et al. (2024); Liu et al. (2025b); Mei et al. (2025); Yang et al. (2025)) from spatial interference between the subpixel display lattice and the camera sampling grid has been widely studied, producing effective learning-based remedies. However, our empirical survey of real-world captures reveals a different, underexplored image degradation that frequently dominates visual quality: ***Flicker-banding***.

Flicker-banding (FB) appears as alternating bright and dark stripes that traverse the image, often approximately horizontal but sometimes tilted or warped. The root cause is temporal: most commodity smartphone sensors employ electronic rolling shutters that expose rows sequentially, while modern displays modulate luminance in time via pulse-width modulation (PWM) or scanning refresh. When the camera's line readout cadence aliases the display's time-varying emission, the temporal mismatch collapses into spatial striping within a single frame. The perceptual impact is so substantial that FB largely distracts attention, obscures screen elements and small fonts, and distorts tone and contrast. FB always makes photos captured unusable for documentation or sharing.

Removing FB is an inherent challenge for three reasons. (i) ***Missing screen-side metadata.*** During photography, the camera has no access to the screen's driving parameters (*e.g.*, PWM frequency, duty cycle, scanning order), which are device dependent and mode dependent. Therefore, hardware-side FB mitigation solutions are extremely difficult. (ii) ***Various morphological types.*** Stripe orientation, spacing, duty cycle, and contrast depend jointly on the sensor readout speed, exposure, gain, and display technology and settings. Different parameters yield diverse and non-stationary patterns that strain single-prior restorers. (iii) ***Partial information loss.*** In severe cases, the dark phase of the modulation produces near-black bars that overwrite scene content. Successful restoration must reason about missing structures, not merely denoise or deblur. All these factors differentiate FB from classic moiré or global deflicker and call for a task-specific image reconstruction solution.

We assume that a dedicated learning-based remedy is necessary and feasible. To the best of our knowledge, it is the **first** work that formulates FB removal for single images with neural networks. Firstly, the lack of training data is the central barrier because it is difficult to collect large-scale paired training sets with banding-free references. We therefore design a ***simulation pipeline*** that injects realistic banding into high-quality images in the luminance domain, with stochastic jitter in angle, spacing, and width. Additional feathered transitions and sensor noise are applied for more realistic appearance as well. What's more, for objective evaluation, we collect a paired ***real-world dataset*** by capturing banded observations with short exposure and banding-free references with long exposure from fixed viewpoints and aligning them at the pixel level. Finally, our model, **R**emoval of **I**mage **F**licker-banding via **L**atent diffusion **E**nhancement (***RIFLE***) is proposed. Based on the baseline PiSA-SR (Sun et al. (2025)), we propose two task-aware components to enhance the model performance. (I) ***Flicker-banding Prior Estimator (FPE)*** is proposed to predict banding attributes, and we inject it into the restoration network. (II) We propose ***Masked Loss*** (**ML**) that emphasizes supervision on banded regions while preserving global fidelity. Results of the experiments on our real-world FB dataset indicate RIFLE gains a great advantage over other recent compared methods.

Overall, as shown in Fig. 1, our contributions are listed as follows:

- To the best of our knowledge, it is ***the first work*** to research the simulation and removal of flicker-banding (FB) and establishes a strong foundation for subsequent research.

- We propose a simulation pipeline for FB to build a large-scale training dataset and a real-world testing dataset for evaluating the FB removal model's performance.

- We present ***RIFLE***, a one-step diffusion-based model with our proposed Flicker-banding Prior Estimator (***FPE***) and Masked Loss (***ML***) tailored to the FB artifacts.

- Our RIFLE achieves substantial gains over other recent image reconstruction methods in both quantitative metrics and visual comparisons. RIFLE is capable of addressing various FB patterns and can be directly applied to many real-world scenarios.

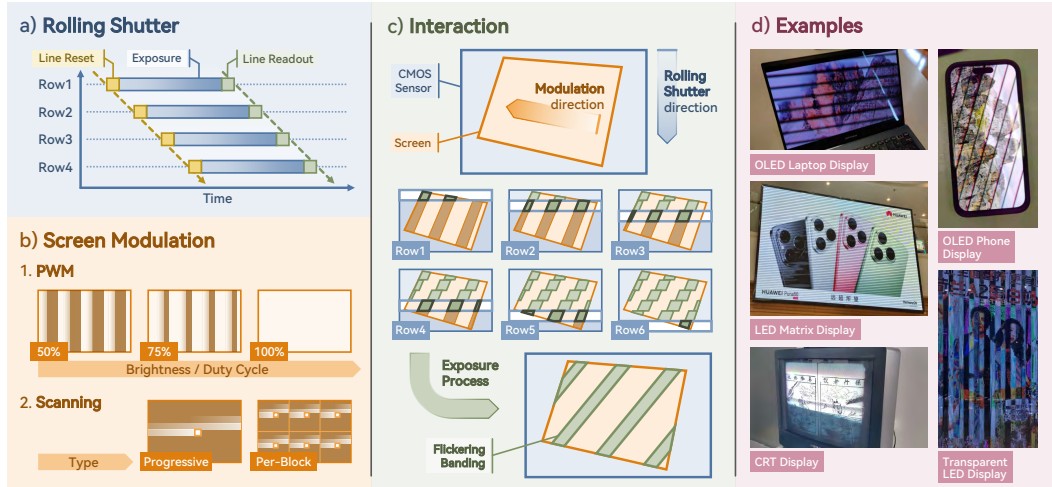

Figure 2: Flicker-banding when filming screens with smartphone cameras. **a)** Rolling shutter exposure process. **b)** Typical display brightness modulation (e.g., PWM and scanning refresh). **c)** Interaction between camera exposure and screen modulation leads to banding artifacts. **d)** Example banding patterns captured from different display technologies.

## 2 RELATED WORK

Researchers have addressed display-camera artifacts, such as moiré patterns. Many works (Sun et al. (2018); Yu et al. (2022); Xu et al. (2024); Liu et al. (2025b)) make great progress in constructing moiré datasets and designing demoiré models. Other research on rolling-shutter degradation spans physics-based and learning-based correction, including joint rolling-shutter correction and deblurring (Zhong et al. (2021); Cao et al. (2024)). Parallel efforts on flickering artifacts target fluctuations caused by temporal variations in global illumination such as fluorescent-light flicker (Köhler et al. (2021); Lin et al. (2023)) using typical methods like CycleGAN (Zhu et al. (2017)).

However, prior studies have not modeled or restored the stripe-like flicker banding that arises in rolling-shutter smartphone imaging of PWM- or scan-driven displays, and no paired datasets are available. Our work releases the first dataset for rigorous evaluation and formulates flicker-banding restoration based on diffusion models (Ho et al. (2020); Xia et al. (2023); Rombach et al. (2022)).

## 3 PRELIMINARIES

**Flicker-Banding (FB).** When recording OLED or LED matrix displays with a smartphone camera, periodic bright and dark stripes often appear across the image. These FB artifacts arise from temporal mismatch between the camera's acquisition process and the display's brightness modulation.

Most smartphone cameras use Complementary Metal-Oxide Semiconductor (CMOS) sensors, with an electronic rolling shutter mechanism (Durini (2019)). This mechanism exposes and reads out each row of the photodiode array one by one, introducing small temporal offsets across the frame, making the captured signal sensitive to time-varying illumination.

OLED-based displays typically regulate brightness through pulse-width modulation (PWM) (Geffroy et al. (2006)), while LED matrix displays often use a scanning refresh scheme. In both cases, only a subset of pixels are lit simultaneously, creating high-frequency temporal fluctuations.

The FB effect appears when the sequential exposure process overlaps with the screen's modulated emission (Sumner (2020)), as shown in Fig. 2. This temporal aliasing projects invisible temporal dynamics into visible spatial patterns, resulting in striping artifacts.

The visibility and morphology of FB are influenced by both camera (e.g., exposure time, line readout speed) and display factors, resulting in a variety of patterns. Additional details on display techniques, modulation methods, and FB patterns are referred to section A of the supplementary materials.

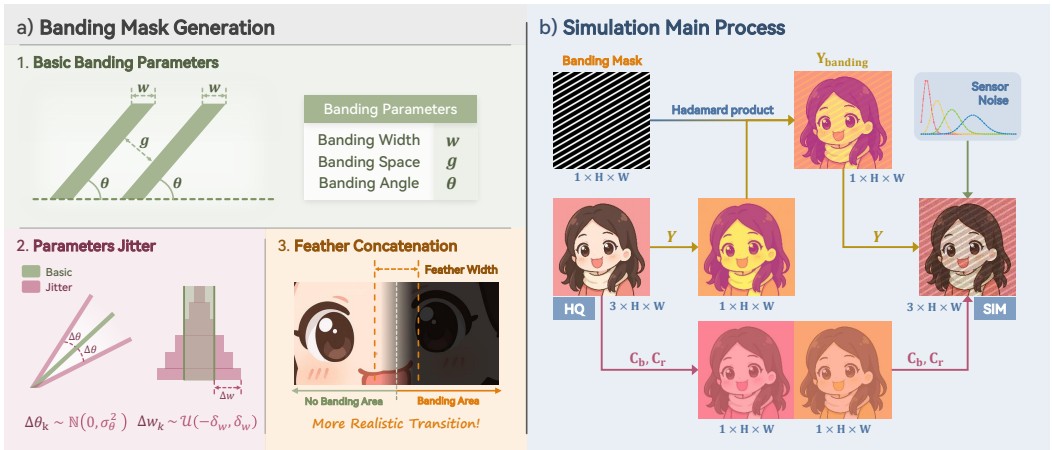

Figure 3: Our flicker-banding simulation pipeline design. **a)** Stage 1: We generate the general framework based on the basic banding parameters and introduce parameter jitter and feather concatenation for a more realistic transition. **b)** Stage 2: We overlay the flicker-banding mask on the Y-channel of high-quality (HQ) images and add sensor noise to the reconstructed images.

## 4 METHODS

### 4.1 FLICKER-BANDING DATASET PIPELINE

To our knowledge, there are no existing datasets of flicker-banding (FB) degradations. To address the problems that severe visual discomforts brought by FB when taking photos, it is essential to construct a dataset composed of various types of FB for training and testing. Therefore, we propose a Flicker-Banding simulation pipeline in Fig. 3 for training, and a Real-World dataset for testing.

#### 4.1.1 SIMULATED FLICKER-BANDING DATASETS FOR TRAINING DATASETS

Considering that paired Real-World FB images or videos are difficult to obtain and the dataset volume for training is enormous, we consider simulation as a feasible solution.

Let the high-quality (HQ) RGB image be $I_{HQ} \in [0,1]^{3 \times H \times W}$ with pixel coordinates $(x, y)$. We form a stripe-aligned local coordinate $(u, v)$ by rotating pixel coordinates through banding angle $\theta$:

$$\begin{bmatrix} u \\ v \end{bmatrix} = \begin{bmatrix} \cos\theta & \sin\theta \\ -\sin\theta & \cos\theta \end{bmatrix} \begin{bmatrix} x - \frac{W-1}{2} \\ y - \frac{H-1}{2} \end{bmatrix}, \qquad \theta \in [-\pi, \pi). \tag{1}$$

Given nominal stripe width $w > 0$ and gap $g > 0$, we get the spatial period $P = w + g$. With a normal-direction phase offset $\phi$ (in pixels), the centerline of the $k$-th stripe is

$$L_k^c(u) = kP + \phi, \qquad k \in \mathbb{Z}. \tag{2}$$

The basic FB mask (1 indicates banding area, 0 indicates non-banding area ) is

$$\mathcal{M}(u, v) = \begin{cases} 1, & \exists k : |v - L_k^c(u)| \leq \frac{w_0}{2}, \\ 0, & \text{otherwise.} \end{cases} \tag{3}$$

To approximate realistic non-ideal flicker-banding, we introduce jitter to the orientation angle, the inter–stripe spacing and width, and the edges along the stripe axis.

**Orientation angle jitter.** To allow realistic departures from the nominal orientation $\theta$, we model the local orientation of the $k$-th stripe along its tangential coordinate $u$ as

$$\theta_k(u) = \theta_0 + \Delta\theta_k(u), \tag{4}$$

where $\triangle\theta_k(u) \sim \mathcal{N}(0, \sigma_\theta{}^2)$ is a zero-mean Gaussian perturbation with the variance $\sigma_\theta^2$.

**Spacing and width fluctuation.** We denote the $k$-th stripe centerline with spacing jitter as

$$L_k^c(u) = kP + \phi + \Delta g_k, \qquad \Delta_g \sim \mathcal{U}(-\delta_g, \delta_g), \tag{5}$$

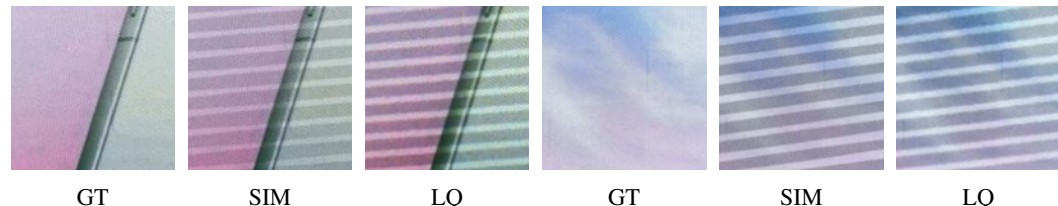

GT      SIM      LQ      GT      SIM      LQ

Figure 4: Visual comparison between our simulated flicker-banding and real-world flicker-banding. GT indicates the real-world non-banding images, on which our simulation pipeline is conducted. SIM indicates our simulation FB images, while LQ indicates real-world FB images.

where $\delta_g$ is the spacing-jitter amplitude. The banding width jitter with width-jitter amplitude $\delta_w$ is

$$w_k = w + \Delta w_k, \qquad \Delta_w \sim \mathcal{U}(-\delta_w, \delta_w). \tag{6}$$

**Axial edge fluctuation.** The top and bottom edges meander along the stripe axis :

$$\begin{cases} v_k^{\text{top}}(u) = v_k^{\text{c}}(u) + \frac{1}{2} w_k(u) + \delta_{\text{edge}}\, \eta_{\text{top}}(u) \\ v_k^{\text{bot}}(u) = v_k^{\text{c}}(u) - \frac{1}{2} w_k(u) + \delta_{\text{edge}}\, \eta_{\text{bot}}(u) \end{cases}, \tag{7}$$

where $\eta_{\text{top}}, \eta_{\text{bot}}$ are low-pass 1D random processes and $\delta_{edge}$ sets the normal jitter amplitude.

We convert HQ image $I_{HQ}$ from RGB space to $\text{YC}_b\text{C}_r$ space and isolate luminance channel:

$$I_{HQ}^Y = K_R I_{HQ}^R + K_G I_{HQ}^G + K_B I_{HQ}^B, \tag{8}$$

where $K_R = 0.299, K_B = 0.114, K_G = 1 - K_R - K_B = 0.587$. It is worth noting that $I^C \in [0,1]^{1 \times H \times W}$ $(C = R, G, B, Y, C_b, C_r)$ indicates the $C$ channel of the image $I$.

Then we apply the banding only on the luminance channel with darkening factor $v_Y \in (0, 1]$:

$$I_{LQ}^Y = \underbrace{v_Y I_{HQ}^Y \mathcal{M}}_{Banding\ Area} + \underbrace{I_{HQ}^Y (1 - \mathcal{M})}_{Nonbanding\ Area}. \tag{9}$$

The reason for selecting luminance as the sole target for the mask is provided in the section C of the supplementary material, which indicates the real-world FB mainly relies on the luminance channel.

We can obtain the simulated FB image $I_{LQ}$ by overlaying the processed channels and incorporating a sensor-noise term to emulate the charge non-uniformity induced by short exposure times:

$$I_{LQ} = \mathcal{C}(I_{LQ}^Y, I_{HQ}^{C_b}, I_{HQ}^{C_r}) + \zeta(\mathcal{C}(I_{LQ}^Y, I_{HQ}^{C_b}, I_{HQ}^{C_r})), \tag{10}$$

where $\zeta$ indicates the sensor noise with Poisson noise strength $\alpha$ and Gaussian noise strength $\sigma_r^2$:

$$\zeta(I) = \sqrt{\alpha I + \sigma_r^2}\, \varepsilon, \qquad \varepsilon \sim \mathcal{N}(0, 1). \tag{11}$$

Notably, we apply feathered blending at stripe boundaries to produce smoother transitions, yielding more natural visual effects that better approximate real-world behavior. We provide the visual comparison between our simulated flicker-banding and real-world flicker-banding in Fig. 4. Obviously, we achieve a remarkably close visual effects, which is crucial for the validity of subsequent models.

### 4.1.2 REAL-WORLD FLICKER-BANDING DATASETS FOR TESTING DATASETS

Although our simulation pipeline yields ample training samples, a real-world benchmark is essential for objective evaluation. To this end, we collected an evaluation set comprising five scenes containing electronic displays. For each scene, we captured paired images from a fixed viewpoint: a flicker-banding observation using a short exposure (fast shutter) and a banding-free reference using a long exposure (slow shutter). All pairs were registered at the pixel level. After preprocessing and quality control, the dataset contains 105 image pairs at a native resolution of 4096×3072. Because full-frame comparisons exhibited limited discriminative power (global metrics tended to saturate), we localized the evaluation to screen regions. Specifically, we used SAM2 (Ravi et al. (2024)) to delineate screen masks and then cropped 424 paired patches of 512×512 size from within segmented screen regions to assess the debanding methods presented in the subsequent sections.

Figure 5: Overview of our model design. **a)** We train a banding prior estimator (BPE) to predict the banding parameters of low-quality (LQ) inputs. **b)** We introduce the pretrained BPE to the diffusion structure, resulting in more banding priors for the model. **c)** We propose a masked loss (ML) to guide the model to focus more on the reconstruction of the image content in the banding area.

## 4.2 Design of Flicker-Banding Removing Model

Although flicker-banding (FB) is frequently encountered in our daily photography, especially when shooting the electronic screens. However, effective researches or technical solutions remain limited. Due to uncertainties in the screen's scanning mechanisms and material characteristics, the hardware-side solutions are difficult to engineer, making it challenging to modify capture devices to avoid banding. Therefore, we adopt an image restoration model in Fig. 5 to reconstruct images affected by FB and thereby enhance customers' photography experience. We select one-step diffusion model, PiSA-SR(Sun et al. (2025)), as our baseline model for its high efficiency and great performance.

### 4.2.1 Loss Design of Removing Model

We assume the low-quality (LQ), prediction, and ground truth (GT) are $x_{\text{LQ}}$, $x_{\text{Pred}}$, and $x_{\text{GT}} \in \mathbb{R}^{B \times C \times H \times W}$, and the mask generated by simulation process is $\mathcal{M} \in [0,1]^{B \times 1 \times H \times W}$. It is worth noting that 1 denotes banding area while 0 denotes background area. In the removal of FB images, the background area of $x_{\text{LQ}}$ is highly similar to that of $x_{\text{GT}}$. Therefore, we need to pay more attention to the reconstruction of the banding area. We apply $\lambda_{\text{banding}} \in [0,1]$ to balance the background and banding regions. Also, $\lambda_{\text{Pixel}}, \lambda_{\text{Perceptual}} \geq 0$ are introduced to weight different loss terms.

**Area-decoupled masked mean operator.** Let $I \in \mathbb{R}^{B \times C \times H \times W}$ be a three-channel image tensor and $\mathcal{M} \in [0,1]^{B \times 1 \times H \times W}$ be a single-channel nonnegative weight map. Let $\tilde{\mathcal{M}}$ denote $\mathcal{M}$ broadcast to the shape of $I$. We define the masked mean operator $\langle\!\langle \cdot \rangle\!\rangle$ as:

$$\langle\!\langle I \rangle\!\rangle_{\mathcal{M}} = \frac{\sum I \odot \tilde{\mathcal{M}}}{\sum_{b,c,h,w} \mathbf{1} \odot \tilde{\mathcal{M}} + \varepsilon}, \tag{12}$$

where $\varepsilon > 0$ ensures numerical stability and $\sum$ indicates the element-wise summation operation.

**Masked Pixel Loss.** We apply mean squared error (MSE) to guide pixel-level reconstruction. We compute the MSE matrix with the model prediction output $x_{\text{Pred}}$ and the ground-truth $x_{\text{GT}}$ as:

$$\mathcal{L}_{\text{Pixel}} = (x_{\text{Pred}} - x_{\text{GT}})^2, \tag{13}$$

In order to guide the model to pay more attention to restoring the image content in the banding areas, we apply the banding mask $\mathcal{M}$ on $\mathcal{L}_{\text{Pixel}}$ to obtain the masked MSE loss as:

$$\mathcal{L}_{\text{Pixel}}^{\text{Masked}} = \lambda_{\text{banding}} \langle\!\langle \mathcal{L}_{\text{Pixel}} \rangle\!\rangle_{\mathcal{M}} + (1 - \lambda_{\text{banding}}) \langle\!\langle \mathcal{L}_{\text{Pixel}} \rangle\!\rangle_{1-\mathcal{M}}. \tag{14}$$

**Masked Perceptual Loss.** We apply a great image quality assessment method, LPIPS (Zhang et al. (2018)), to enhance the quality of the overall reconstructed image. The LPIPS network $\Phi$ produces a per-pixel perceptual distance map for the inputs normalized to $[-1, 1]$, $\hat{x_{\text{Pred}}}$ and $\hat{x_{\text{GT}}}$. as follows:

$$\mathcal{L}_{\text{Perceptual}} = \Phi(\hat{x_{\text{Pred}}}, \hat{x_{\text{GT}}}) \in \mathbb{R}^{B \times 1 \times h \times w}, \tag{15}$$

Similarly, we apply the banding mask $\mathcal{M}$ on $\mathcal{L}_{\text{Perceptual}}$ to obtain the masked perceptual loss:

$$\mathcal{L}_{\text{Perceptual}}^{\text{Masked}} = \lambda_{\text{banding}} \langle\!\langle \mathcal{L}_{\text{Perceptual}} \rangle\!\rangle_{\mathcal{M}} + (1 - \lambda_{\text{banding}}) \langle\!\langle \mathcal{L}_{\text{Perceptual}} \rangle\!\rangle_{1-\mathcal{M}}. \tag{16}$$

**Merged Masked Loss.** To achieve the balance of the pixel-level and overall quality of the reconstructed image $x_{\text{Pred}}$, we can obtain the merged masked loss $\mathcal{L}$ as follows:

$$\mathcal{L} = \lambda_{\text{Pixel}} \mathcal{L}_{\text{Pixel}}^{\text{Masked}} + \lambda_{\text{Perceptual}} \mathcal{L}_{\text{Perceptual}}^{\text{Masked}}. \tag{17}$$

### 4.2.2 FLICKER-BANDING PRIOR ESTIMATOR

Inspired by diffusion-based reconstruction methods, we consider incorporating more prior knowledge about FB to guide the reconstruction process. We propose a Flicker-Banding Prior Estimator (FPE) to provide the key banding prior for diffusion model to enhance model performance.

A key advantage of the simulated dataset is that it provides accurate pixel-level annotations of the banding parameters (*e.g.*, banding width $w$, banding spacing $g$, and banding angle $\theta$), which are relatively difficult to obtain from the real-world dataset. We feed the simulated flicker-banding (FB) images into an estimator that predicts our selected parameters, inverse to the FB simulation process. Our FPE adopts a ResNet-based architecture (He et al. (2016)), which is effective while introducing minimal additional computational overhead to the overall diffusion model.

We introduce the pre-trained FPE to our baseline model structure, and concatenate the FPE and the UNet with an embedding module. In the training process, we fine-tune the UNet with LoRA, and perform full-parameter fine-tuning on the embedding module, freezing the VAE and BPE.

## 5 EXPERIMENTS

### 5.1 EXPERIMENTS SETUP

**Data Construction.** For the training datasets, we employ our proposed simulation pipeline on LSDIR (Li et al. (2023)) and UHDM (Yu et al. (2022)). LSDIR is a large-scale super-resolution dataset while UHDM is an outstanding demoireing dataset. Both of them have a large amount of high-quality images, and we utilize them to generate flicker-banding images. Considering the specific scenarios of flicker-banding occurrence, we assume that UHDM can better represent images of screens captured by cameras, thereby enhancing the model's understanding of screen-shooting scenarios. LSDIR corresponds to more general scenarios, improving model's generalization ability. For the testing datasets, we use our proposed real-world datasets, consisting of 424 paired patches of $512 \times 512$ size. Results from real-world datasets better demonstrate the model's practical value.

**Evaluation Metrics.** We employ reference-based evaluation metrics, including PSNR, SSIM (Wang et al. (2004)), LPIPS (Zhang et al. (2018)), DISTS (Ding et al. (2020)), FSIM (Zhang et al. (2011)), and GMSD (Xue et al. (2014)). Non-reference evaluation metrics are excluded from our evaluation process, as they often yield similar scores for banding and banding-free images. They can't recognize the flicker-banding and fail to provide a reliable assessment of model performance.

**Implementation Details.** For LoRA finetuning of the diffusion model, we set the rank to 32 with a learning rate of $5 \times 10^{-5}$. The training process is performed using images of resolution $512 \times 512$. We set the training batch size of our model to 4, consuming about 43.3 GB of GPU memory and a complete training duration of approximately 25.8 hours for 50000 iterations. For the masked loss, we assign the weights $\lambda_{\text{banding}} = 0.8$, $\lambda_{\text{Pixel}} = 1.0$, and $\lambda_{\text{Perceptual}} = 2.0$.

**Compared Methods.** Owing to the lack of researches on the flicker-banding, we have to compare our methods with recent image reconstruction methods in other tasks. To ensure the fair comparison, we finetune the compared methods with our simulated dataset as well. We adopt MAT (Xie et al. (2025)) as the representative method for transformer-based approaches. InvSR (Yue et al. (2025)) and PiSA-SR (Sun et al. (2025)) are representative of diffusion-based methods. Step1X (Liu et al. (2025a)) stands for image-editing models, which can also solve lots of problems in low-level vision.

Table 1: Quantitative experiments results of different debanding methods on **cropped** real-world flicker-banding datasets. All models are finetuned with simulated datasets. The best and second best results are colored with red and blue. RIFLE gains a significant advantage over other methods.

| Methods | PSNR↑ | SSIM↑ | ms-SSIM↑ | LPIPS↓ | DISTS↓ | FSIM↑ | GMSD↓ |
|---|---|---|---|---|---|---|---|
| LQ | 19.43 | 0.5636 | 0.6364 | 0.3374 | 0.2213 | 0.7907 | 0.2091 |
| MAT (Xie et al. (2025)) | 20.28 | 0.5984 | 0.7078 | 0.2967 | 0.2082 | 0.8214 | 0.1804 |
| InvSR (Yue et al. (2025)) | 19.08 | 0.5260 | 0.6328 | 0.4367 | 0.2801 | 0.7362 | 0.2177 |
| PiSA-SR (Sun et al. (2025)) | 20.57 | 0.6264 | 0.8056 | 0.2389 | 0.1732 | 0.8663 | 0.1457 |
| Step1X (Liu et al. (2025a)) | 19.20 | 0.5619 | 0.6317 | 0.3487 | 0.2249 | 0.7867 | 0.2114 |
| RIFLE (ours) | 20.66 | 0.6220 | 0.8067 | 0.2456 | 0.1723 | 0.8711 | 0.1433 |

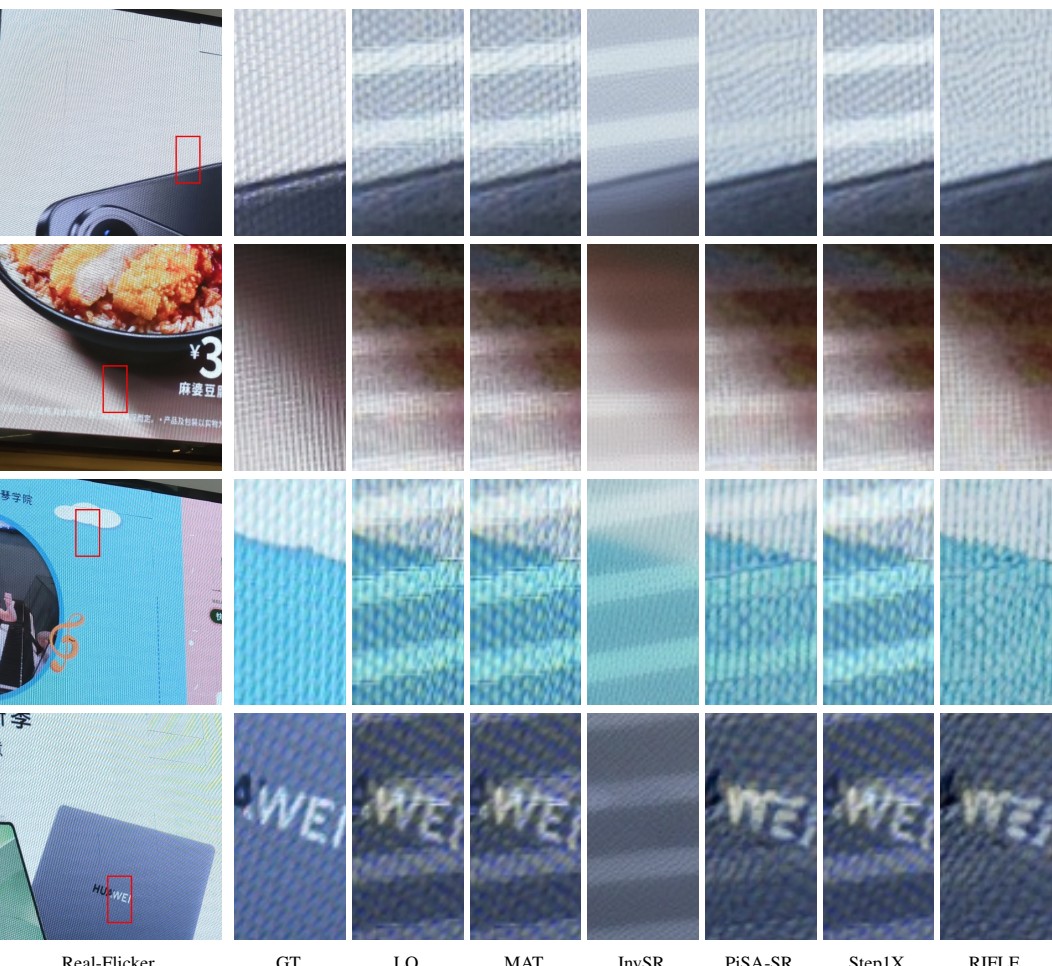

| Real-Flicker | GT | LQ | MAT | InvSR | PiSA-SR | Step1X | RIFLE |

Figure 6: Visual comparison with flicker-banding images (LQ), banding-free images (GT), and other debanding methods on our Real-Flicker dataset. RIFLE gains great advantages over other methods.

## 5.2 MAIN RESULTS

**Quantitative Results.** We provide the quantitative experimental results of different methods on our real-world dataset in Tab. 1. Recent competing methods generally show limited effectiveness in addressing flicker-banding (FB) artifacts. It is obvious that our RIFLE holds an advantage over other methods on most metrics. We discovered an interesting phenomenon that even the raw inputs can obtain relatively high scores on various reference-based metrics. The advantage of our method will be further discussed in the following visual comparison. Although they are able to quantify the discrepancy between model outputs and the ground-truth (GT) banding-free images, they are largely insensitive to FB artifacts. We assume that the phenomenon is reasonable because banding entails minimal loss of fine details, and non-banding regions of FB are highly similar to those of GT.

Table 2: Ablation study results on **full-sized** real-world flicker-banding datasets. ML indicates masked loss, FPE indicates the flicker-banding prior estimator, and ML+FPE indicates the whole RIFLE model. The best and second best results in the same setting are colored with red and blue.

| Methods | PSNR↑ | SSIM↑ | ms-SSIM↑ | LPIPS↓ | DISTS↓ | FSIM↑ | GMSD↓ |
|---------|-------|-------|----------|--------|--------|-------|-------|
| LQ | 21.78 | 0.7373 | 0.7655 | 0.2322 | 0.1219 | 0.8594 | 0.1848 |
| Baseline | 22.12 | 0.7490 | 0.8677 | 0.1812 | 0.0920 | 0.9406 | 0.1297 |
| ML | 22.28 | 0.7505 | 0.8590 | 0.1955 | 0.0984 | 0.9372 | 0.1344 |
| FPE | 22.15 | 0.7349 | 0.8683 | 0.1910 | 0.0918 | 0.9431 | 0.1337 |
| ML+FPE | 22.30 | 0.7425 | 0.8697 | 0.1902 | 0.0908 | 0.9460 | 0.1286 |

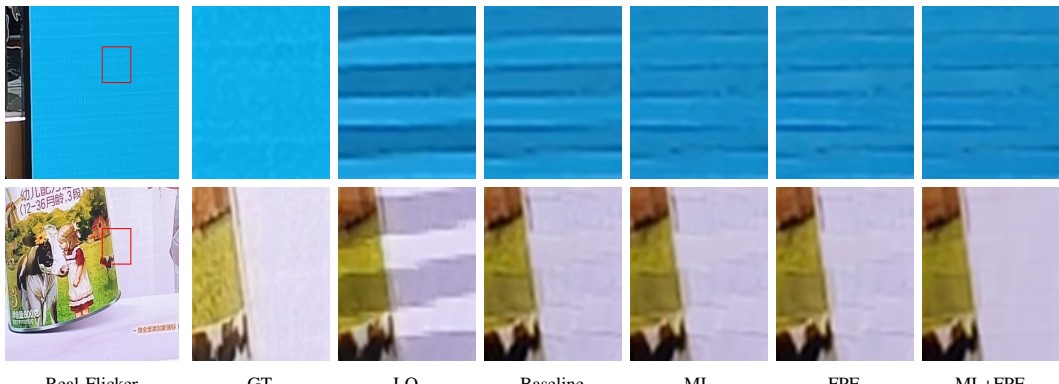

| Real-Flicker | GT | LQ | Baseline | ML | FPE | ML+FPE |

Figure 7: Visual comparison of the ablation study experiments on our Real-Flicker dataset.

**Visual Comparison.** We compare the visual performance of our method with recent image reconstruction approaches, and present the results in Figs. 6. Despite being trained on our simulated dataset, the competing methods still struggle to handle flicker-banding (FB) artifacts. Conspicuous residual stripe patterns remain in their processed results, degrading perceived visual quality. In contrast, RIFLE effectively handles FB artifacts in real-world scenes. Under mild degradation, it nearly eliminates the stripes while maintaining high fidelity to the original image. Under severe degradation, it still removes the majority of stripes with only minimal residuals, whereas compared methods offer virtually no improvement in heavy-banding cases. Our baseline method, PiSA-SR (Sun et al. (2025)), also performs relatively well after being finetuned with our simulated dataset. However, owing to the introduction of our proposed components, RIFLE eliminates more stripe artifacts while maintaining higher consistency with the ground truth (GT), as shown in Fig. 6.

### 5.3 ABLATION STUDY

We conduct an ablation on our proposed two components: masked loss (ML) and the flicker-banding prior estimator (FPE). The quantitative results are presented in Tab. 2 and the visual comparison in Fig. 7. The baseline leaves noticeable stripe residues, whereas ML focuses learning on banded regions, yielding cleaner outputs and better structural fidelity. FPE introduces an explicit prior that suppresses periodic stripes more aggressively, at times slightly softening textures when used alone. Combining ML and FPE, it removes the most banding with minimal residuals while preserving edges and fine details, leading to consistently stronger results across fidelity, structure-aware, and perceptual criteria as well as clearer visual comparisons, especially under heavier degradation.

### 6 CONCLUSION

In this paper, we propose RIFLE, a diffusion-based framework for removing real-world flicker-banding (FB), together with a simulation pipeline and a paired real-world benchmark. RIFLE couples a flicker-banding prior estimator with a region-focused masked loss to target stripe artifacts while preserving fine details. On our real-world FB datasets, it consistently reduces FB more effectively than recent reconstruction baselines, as confirmed by both quantitative metrics and visual comparisons. Ablations show the two components are effective. To the best of our knowledge, this is the first academic work to tackle the removal of FB artifacts with neural networks. We also provide effective solutions for training and test datasets, laying a solid foundation for subsequent research.

ETHICS STATEMENT

The research conducted in the paper conforms, in every respect, with the ICLR Code of Ethics.

REPRODUCIBILITY STATEMENT

We have provided implementation details in Sec. 5.1. We will also release all the code and models.

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
