# RIFLE: REMOVAL OF IMAGE FLICKER-BANDING VIA LATENT DIFFUSION ENHANCEMENT

# SUPPLEMENTARY MATERIAL

## CONTENTS

## A  FLICKER-BANDING PATTERNS ACROSS DISPLAY TECHNOLOGIES

Despite the commonly observed flicker-banding artifacts shown in section 3 and simulate in section 4.1.1, the specific banding patterns can vary significantly across different display technologies. It's worth noting that various complex stripe types described below are all captured under extreme conditions (very short exposure times). However, extremely severe FB degenerate into dark straight one as the exposure time increases. In everyday photography, we don't use excessively low exposure time and the straight dark FB shown in our dataset is a much more common scene.

### A.1  LED MATRIX DISPLAYS

LED matrix displays often use a scanning refresh scheme, where rows or columns of pixels are activated in succession. The matrix is often refreshed in blocks or chunks of rows, leading to more complex banding patterns when filmed. PWM is also commonly used in LED matrices, further contributing to more complex banding effects.

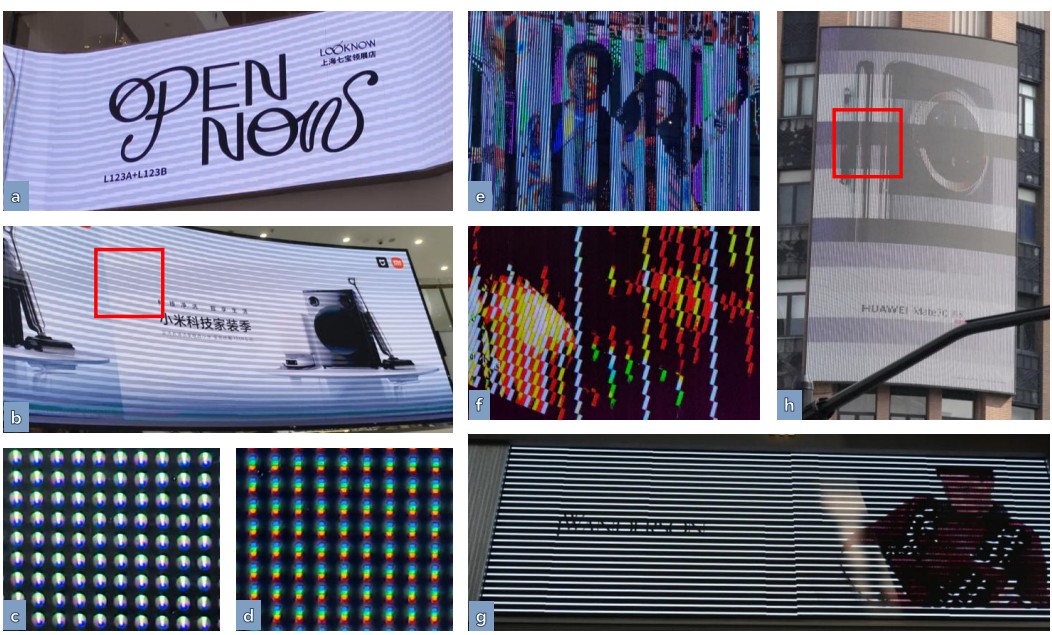

Figure 1: Flicker-banding patterns on LED matrix displays with different scanning refresh mechanisms captured under various conditions. **b)** Thickening of dark stripes, likely due to changes in PWM duty cycle. **c-d)** Different LED subpixel arrangements. **e-f)** Atypical banding patterns. **h)** Complex banding patterns caused by per-block scanning refresh.

### A.2  OLED DISPLAYS

OLED displays typically use a combination of PWM and current modulation to achieve smooth brightness control (Geffroy et al. (2006)). In most cases, the duty cycle is same across the entire panel at a given brightness level, resulting in uniform banding patterns when filmed with a camera. However, some OLED panels may reduce the duty cycle in very dark scenes to enhance contrast, leading to non-uniform banding artifacts. Color shifts may also occur on edges of banding stripes due to phase differences in RGB subpixel driving.

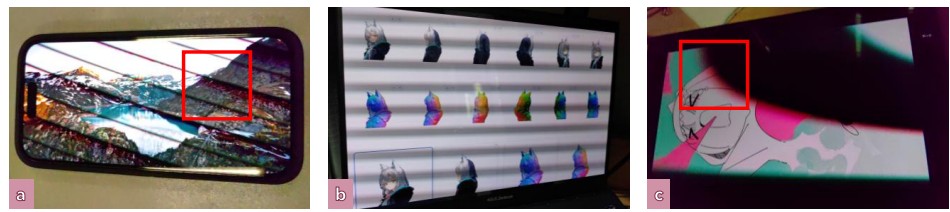

Figure 2: Flicker-banding patterns on OLED displays with PWM driving. **a)** Non-uniform banding in darker areas on an iPhone OLED screen. **b)** Horizontal gray banding on laptop OLED. **c)** Extremely bending and color shifts on banding edges.

### A.3 CRT DISPLAYS

In cathode-ray tube (CRT) monitors, stripe artifacts are primarily related to the scanning process of the electron beam. The resulting banding patterns differ from those in OLED and LED displays, as factors including different scanning methods (e.g., interlaced vs. progressive) and phosphor persistence characteristics come into play.

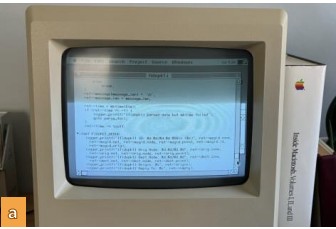 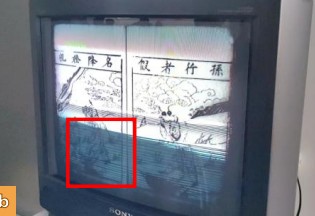

Figure 3: Flicker-banding patterns on CRT monitors with electron beam scanning. **b)** Phosphor persistence effect causing trailing artifacts, leaving a ghosting effect.

### A.4 PROJECTORS

Projectors, especially those using digital light processing (DLP), can also exhibit banding artifacts when captured on camera (Han et al. (2014)). These systems often employ spinning color wheels to generate full-color images, which can interact with the camera's rolling shutter to produce unique colored banding patterns. Brightness modulation is achieved through rapid micromirror switching, which can introduce non-uniform banding under certain filming conditions.

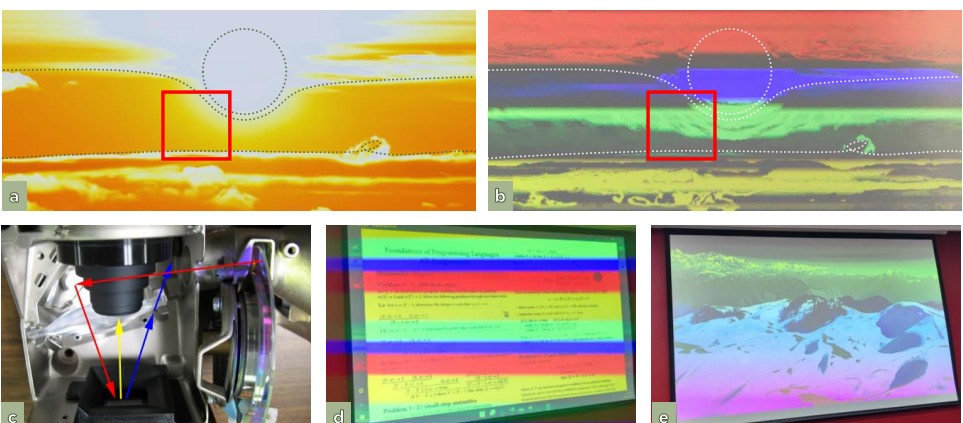

Figure 4: Flicker-banding patterns on DLP projectors with color wheel and micromirror switching. **a-b)** Comparison between original and projected images, highlighting the width of color stripes changes with brightness levels. **c)** Internal components adapted from DMahalko (CC BY-SA 3.0).

Across the surveyed display families, flicker-banding emerges from different timing mechanisms: row/column or per-block scanning and PWM dimming on LED matrix panels, a mixture of PWM and current-driven modulation with subpixel-phase offsets on OLEDs, raster scanning with phosphor persistence on CRTs, and time-sequential illumination (e.g., color wheels) in projectors. These mechanisms create bands that can be straight or curved (due to rolling-shutter readout), globally periodic or piecewise-structured (under per-block scanning/local modulation), and purely luminance-based or chromatic (due to subpixel phase shifts or color wheels).

Such diversity makes restoration challenging: the banding is frequently non-stationary, its frequency and pattern are strongly device-specific, and its appearance can vary with scene content, exposure settings, and viewing angle. Additionally, ISP processing and sensor noise further blur the boundary between true scene detail and artifact. This calls for future work to develop more advanced, physics-informed models and datasets that reflect this variety.

## B    CLARIFICATION ON THE QUANTITATIVE RESULTS

Table 1: Additional ablation study results on **cropped** real-world flicker-banding datasets. ML indicates masked loss, FPE indicates the flicker-banding prior estimator, and ML+FPE indicates the whole RIFLE model. The better results in the same setting are colored with red.

| Image | Methods | PSNR↑ | SSIM↑ | ms-SSIM↑ | LPIPS↓ | DISTS↓ | FSIM↑ | GMSD↓ |
|---|---|---|---|---|---|---|---|---|
| | LQ | 15.39 | 0.6229 | 0.6152 | 0.4208 | 0.1922 | 0.7581 | 0.2524 |
| Banding62_1 | ML | 16.02 | 0.6543 | 0.8372 | 0.3163 | 0.1623 | 0.8413 | 0.1700 |
| | ML+FPE | 16.08 | 0.6701 | 0.8607 | 0.2974 | 0.1559 | 0.8522 | 0.1550 |
| | LQ | 20.79 | 0.4919 | 0.5879 | 0.3720 | 0.1850 | 0.7564 | 0.1865 |
| Banding37_2 | ML | 24.11 | 0.5264 | 0.7646 | 0.2858 | 0.1425 | 0.8793 | 0.1087 |
| | ML+FPE | 23.97 | 0.5226 | 0.7201 | 0.3094 | 0.1566 | 0.8433 | 0.1194 |

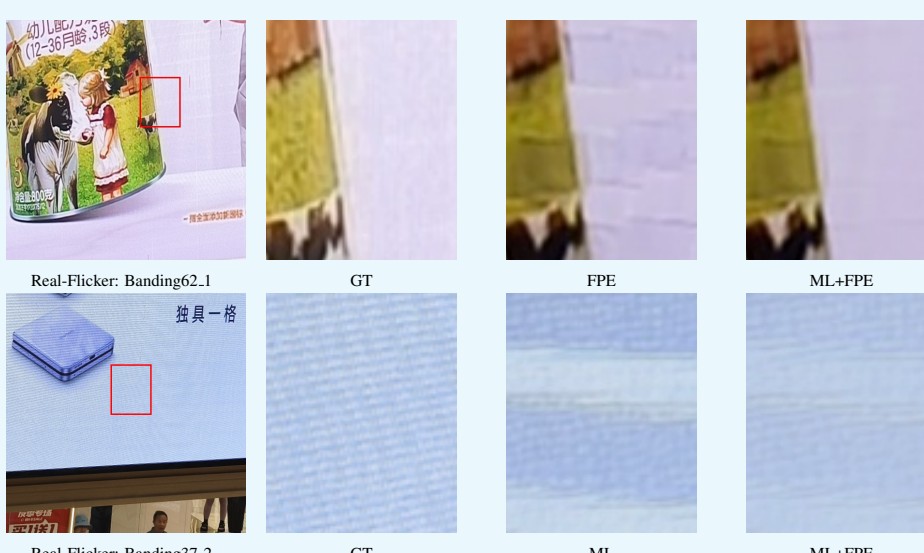

| Real-Flicker: Banding62_1 | GT | FPE | ML+FPE |
|---|---|---|---|
| Real-Flicker: Banding37_2 | GT | ML | ML+FPE |

Figure 5: Additional Visual comparison of ablation study on ML *vs* ML+FPE.

To clarify the saturation of quantitative metrics, we select the comparative experiment, ML *vs* ML+FPE, in ablation study that is the closest in numerical results. Table 1 and Figure 5 provide detailed quantitative results and the visual comparison to describe the connection between the numerical results and the real model performance of removing flicker-banding (FB). For Banding62_1, there are almost no FB residues in the results of ML+FPE, while we are able to observe obvious FB in the results of FPE. According to the visual comparison, ML+FPE should get an obviously higher score of evaluation metrics. However, the numerical results of ML+FPE are slightly superior to ML, which can explain why, even with significant improvements in model performance, the resulting quantitative improvements are often very minor. The quantitative results and visual comparison of Banding37_2 show that quantitative metrics and model performance can also exhibit rebound phenomena. Although this situation is not as common as the one described above, it further reduces the differences in numerical performance between different models across the entire test set.

What's more, even the original FB input (LQ) can obtain a relatively high score in most of the evaluation metrics, indicating that these metrics are not sensitive to the degree of FB. Because these referenced evaluation metrics mainly measure the difference between two images rather than compare the degree of FB directly, many factors like the noise and brightness may have a significant impact on numerical metrics. It further explains that even with great model performance improvement, there are minor enhancement in the quantitative results. Designing a suitable metrics for FB is also one of our future researching directions, but not the focus of this work.

## C  ADDITIONAL COMPARISON EXPERIMENTS

Table 2: Quantitative experiments results of additional debanding methods on **cropped** real-world flicker-banding datasets. All models are finetuned with simulated datasets. The best and second best results are colored with red and blue. RIFLE gains a significant advantage over other methods.

| Methods | PSNR↑ | SSIM↑ | ms-SSIM↑ | LPIPS↓ | DISTS↓ | FSIM↑ | GMSD↓ |
|---|---|---|---|---|---|---|---|
| LQ | 19.43 | 0.5636 | 0.6364 | 0.3374 | 0.2213 | 0.7907 | 0.2091 |
| Notch Filter | 17.86 | 0.4067 | 0.5527 | 0.5340 | 0.3101 | 0.6734 | 0.2172 |
| ESDNet (Yu et al. (2022)) | 20.49 | 0.6077 | 0.7472 | 0.2782 | 0.2075 | 0.8424 | 0.1587 |
| NeRD-Rain (Chen et al. (2024)) | 20.30 | 0.6066 | 0.7447 | 0.2830 | 0.1950 | 0.8462 | 0.1702 |
| RIFLE (ours) | 20.66 | 0.6220 | 0.8067 | 0.2456 | 0.1723 | 0.8711 | 0.1433 |

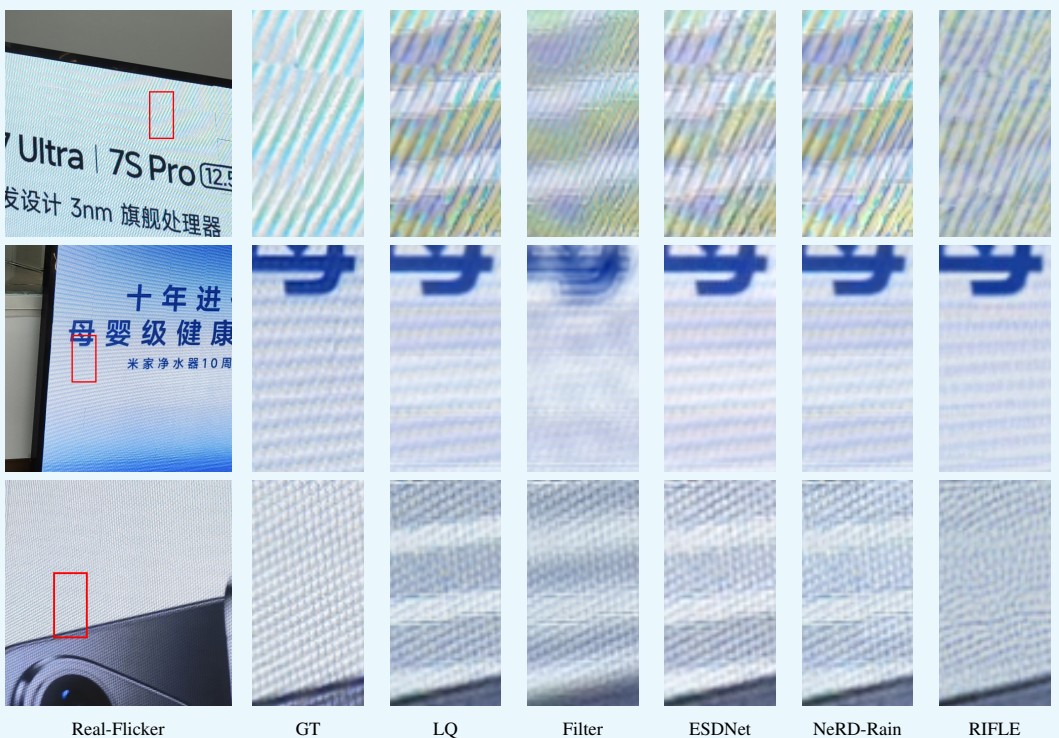

Real-Flicker        GT        LQ        Filter        ESDNet        NeRD-Rain        RIFLE

Figure 6: Visual comparison with FB images (LQ), banding-free images (GT), and additional debanding methods on Real-Flicker dataset. RIFLE still gains great advantages over other methods.

We include additional task-specific baselines in the comparison experiments to demonstrate the superiority of our method. These include the simple notch filter (NF) aligned with our flicker-banding parameter estimator (FPE), as well as ESDNet (Yu et al. (2022)) and NeRD-Rain (Chen et al. (2024)), both of which demonstrate strong performance in their respective tasks.

The NF does not produce substantial improvements and, in fact, degrades image quality. Suppressing flicker in specific frequency bands also results in the loss of original image details at those same frequency locations. The irregular shape of real-world FB can introduce biases in our FPE, making it challenging for the filter to precisely capture the frequency bands where flicker is present.

ESDNet, an excellent demoiréing method, and NeRD-Rain, a state-of-the-art deraining method, are both fine-tuned using our simulated dataset and have some ability to remove FB.

As shown in Tab. 2, RIFLE demonstrates significant improvements over these methods on all metrics. Figure 10 indicates that RIFLE produces the best visual effects, with almost no FB left.

## D  DEMOIRÉ MODEL PERFORMANCE IN THE CASE OF FLICKER-BANDING

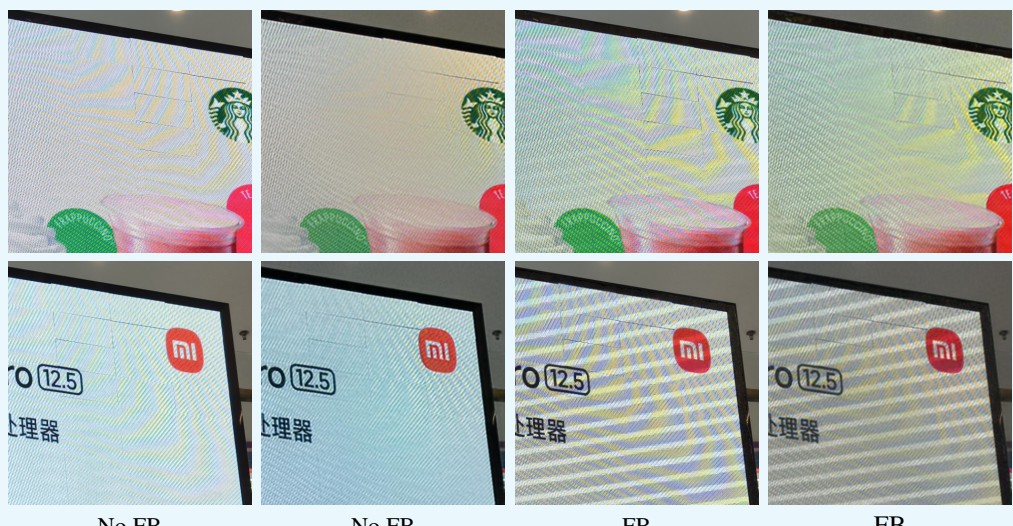

| No FB | No FB | FB | FB |
| before demoiré | after demoiré | before demoiré | after demoiré |

Figure 7: The demoiré performance comparison between no FB scene and FB scene.

In addition to the flikcer-banding (FB), moiré patterns are also bring great damage to the image quality when shooting screen. We find that the performance of demoiré model largely decrease when both of the FB and moiré patterns exist, as shown in Fig. 7. It indicates the difference between FB and moiré patterns and removing them simultaneously is also a valuable issue to be solved.

## E  USER STUDY

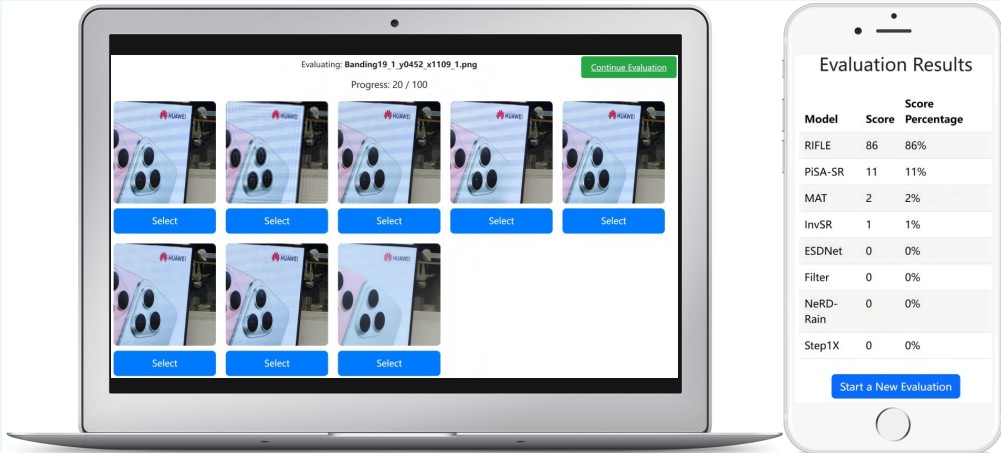

Figure 8: The operating system of the User Study.

Table 3: The results of User Study.

| Methods | RIFLE | PiSA-SR | MAT | InvSR | Step1X | ESDNet | NeRD-Rain | Notch Filter |
|---------|-------|---------|-----|-------|--------|--------|-----------|--------------|
| Score/(%) | 82.4 | 14.7 | 2.7 | 0.02 | 0 | 0.1 | 0.08 | 0 |

Because none of the evaluation metrics can reflect the degree of flicker-banding precisely, a user study is essential to compare different models' performance. Figure 8 shows our user study's operating system, including the evaluation panel and the results panel. The results of the user study are presented in table 3, and our method gains the highest score. The score represents the percentage of rounds in which the model received preference out of the total number of evaluation rounds.

# F   THE OUT-OF-DISTRIBUTION TEST RESULTS.

Input                                                    Reconstructed

Figure 9: The out-of-distribution test results.

We provide the test results on other severe flicker-banding (FB) scenes in Fig. 9, not only on the dark straight FB scenes. The first row indicates that the black stripes are hard to recover and our model only played a minor role in filling in the gaps. The second row indicates that our model makes no difference to the stripes with severe color shifts. The third row implies that there is still a long way to go for our model to reconstruct an image whose details are completely destroyed.

## G  ADDITIONAL VISUAL COMPARISON

We provide additional visual comparison for different methods on our Real-Flicker dataset in Fig. 10. More results demonstrate our RIFLE's outstanding performance compared to recent image reconstruction methods and truly show great application potential in real-world scenarios.

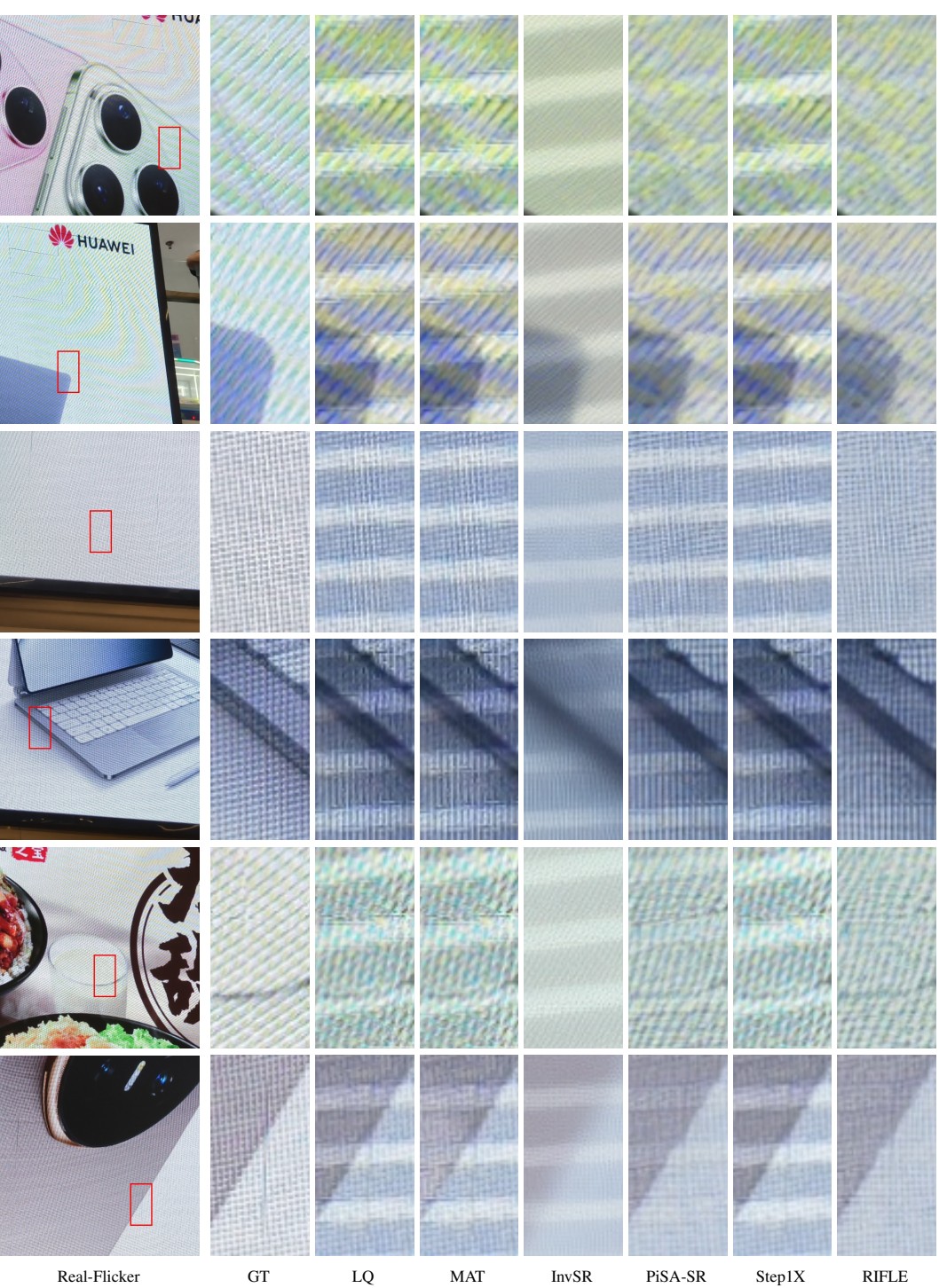

Figure 10: Visual comparison with flicker-banding images (LQ), banding-free images (GT), and other debanding methods on Real-Flicker dataset. RIFLE gains great advantages over other methods.

## H ANALYSIS ON THE FLICKER-BANDING SIMULATION

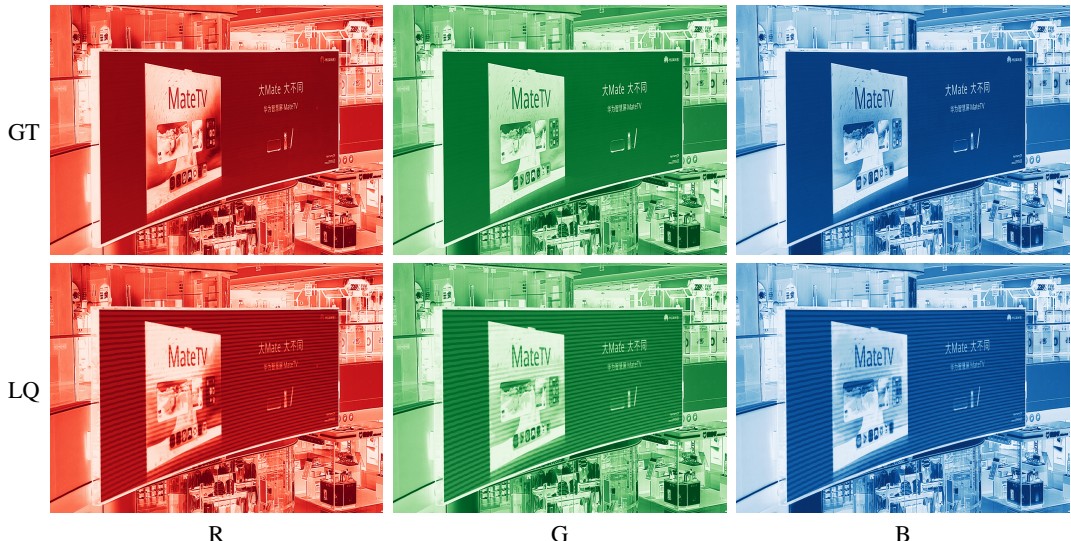

R          G          B

Figure 11: Per-channel visualization of banding-free images (GT) and flicker-banding images (LQ) in RGB color space. Flicker-banding artifacts appear in all three channels of the LQ images.

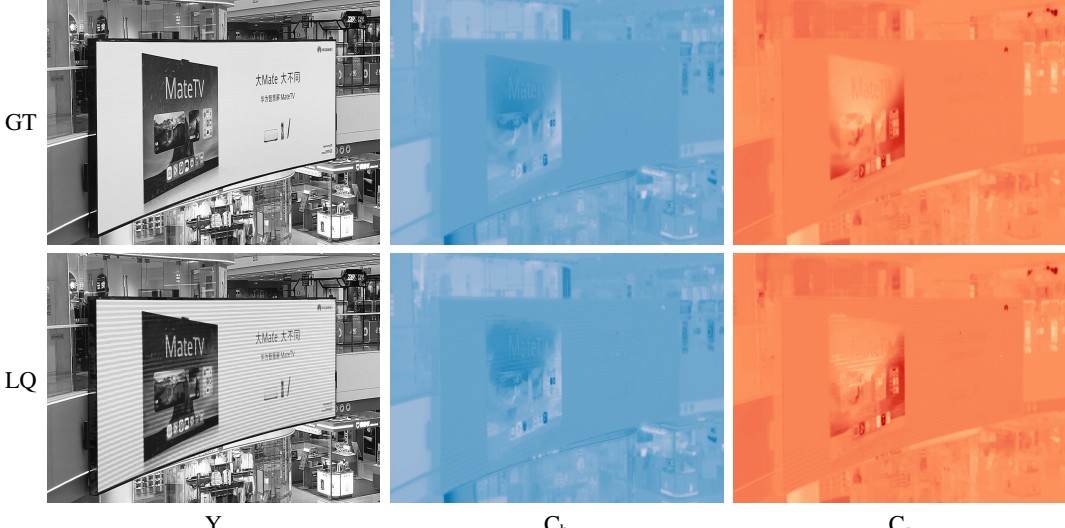

Y          $C_b$          $C_r$

Figure 12: Per-channel visualization of banding-free images (GT) and flicker-banding images (LQ) in $YC_bC_r$ color space. Flicker-banding artifacts only appear in the Y channel of the LQ images.

We analyze each channel of the banding-free image (GT) and the flicker-banding (LQ) and provide the visual comparison in Figs. 11 (RGB) and 12 ($YC_bC_r$). RGB and $YC_bC_r$ color spaces are two common color spaces to express the image construction. We observe pronounced stripe-like flicker-banding artifacts across all three RGB channels of the LQ images, with substantial discrepancies relative to the GT counterparts. This suggests that a faithful simulation of flicker-banding requires overlaying stripe masks simultaneously on all RGB channels.

However, in the $YC_bC_r$ color space, we draw an opposite conclusion to that in the RGB color space. The chrominance channels ($C_b$ and $C_r$) of the LQ images closely match those of the GT images and are visually nearly indistinguishable. More importantly, the LQ images show no flicker-banding in these channels. By contrast, the luminance (Y) channel of the LQ images exhibits pronounced flicker-banding artifacts and deviates substantially from GT.

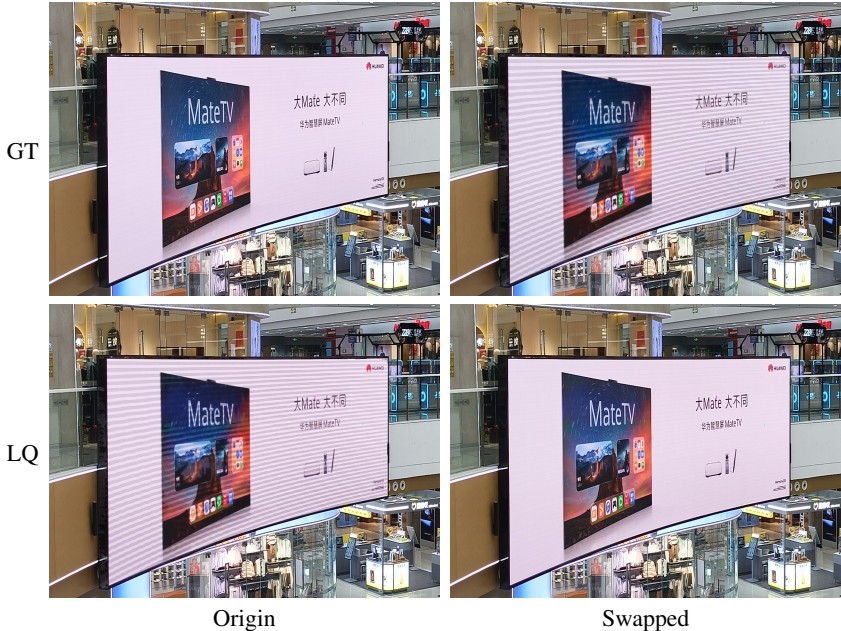

Figure 13: Visualization comparison of Y-channel switching effects. *Origin* shows the original image and *Swapped* shows the image after exchanging its Y channel with the counterpart.

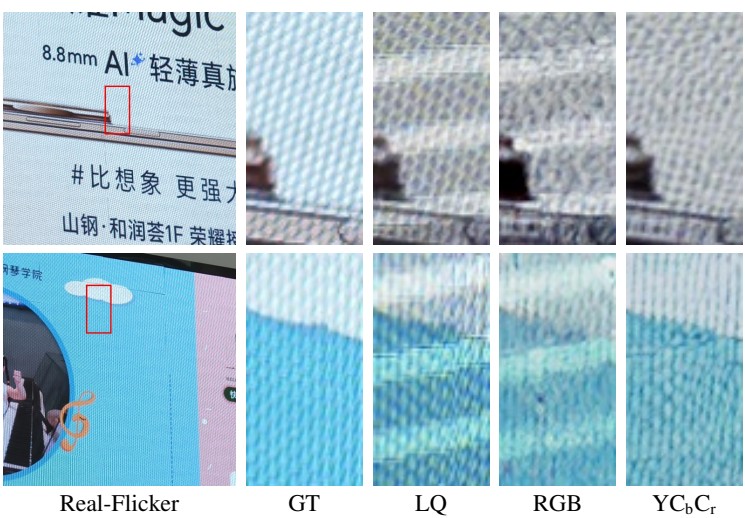

Figure 14: Visual comparison of RIFLE on synthetic datasets based on RGB/$YC_bC_r$ color space.

To further validate this observation, we swapped the Y channels between GT and LQ while keeping their respective Cb and Cr components unchanged. After recomposition (Fig. 13), the LQ image no longer displays visible banding, whereas the GT image exhibits clear flicker-banding. These results confirm that the flicker-banding phenomenon originates primarily in the luminance component (Y) and exerts negligible influence on the chrominance components (Cb and Cr).

Building on this luminance-centric insight, we conducted the experiments in Fig. 14. When a synthetic training set was constructed by superimposing the same mask independently on the R, G, and B channels, the resulting models showed no measurable improvement on real-world data. This outcome underscores that simply applying identical per-channel masks in RGB space is not a faithful approximation of the real phenomenon and introduces a substantial domain gap relative to real-world images. Consequently, a fundamental principle for simulating banding in RGB is that one should avoid superimposing an identical mask across all three channels. Instead, channel-dependent masks must be designed to preserve chrominance while modulating only the luminance (Y).

# I   EXPERIMENTAL RESULTS ON THE SIMULATED DATASETS

We present our experimental results on the simulated datasets in Figs. 15 and 16. The results indicate that MAT, PiSA-SR, and RIFLE achieve great performance of removing flicker-banding artifacts, while other approaches encounter great problems. However, their visual outcomes are largely indistinguishable, implying that the simulated dataset does not constitute the primary bottleneck of their performance. Only the real-world dataset can reflect their performance gap.

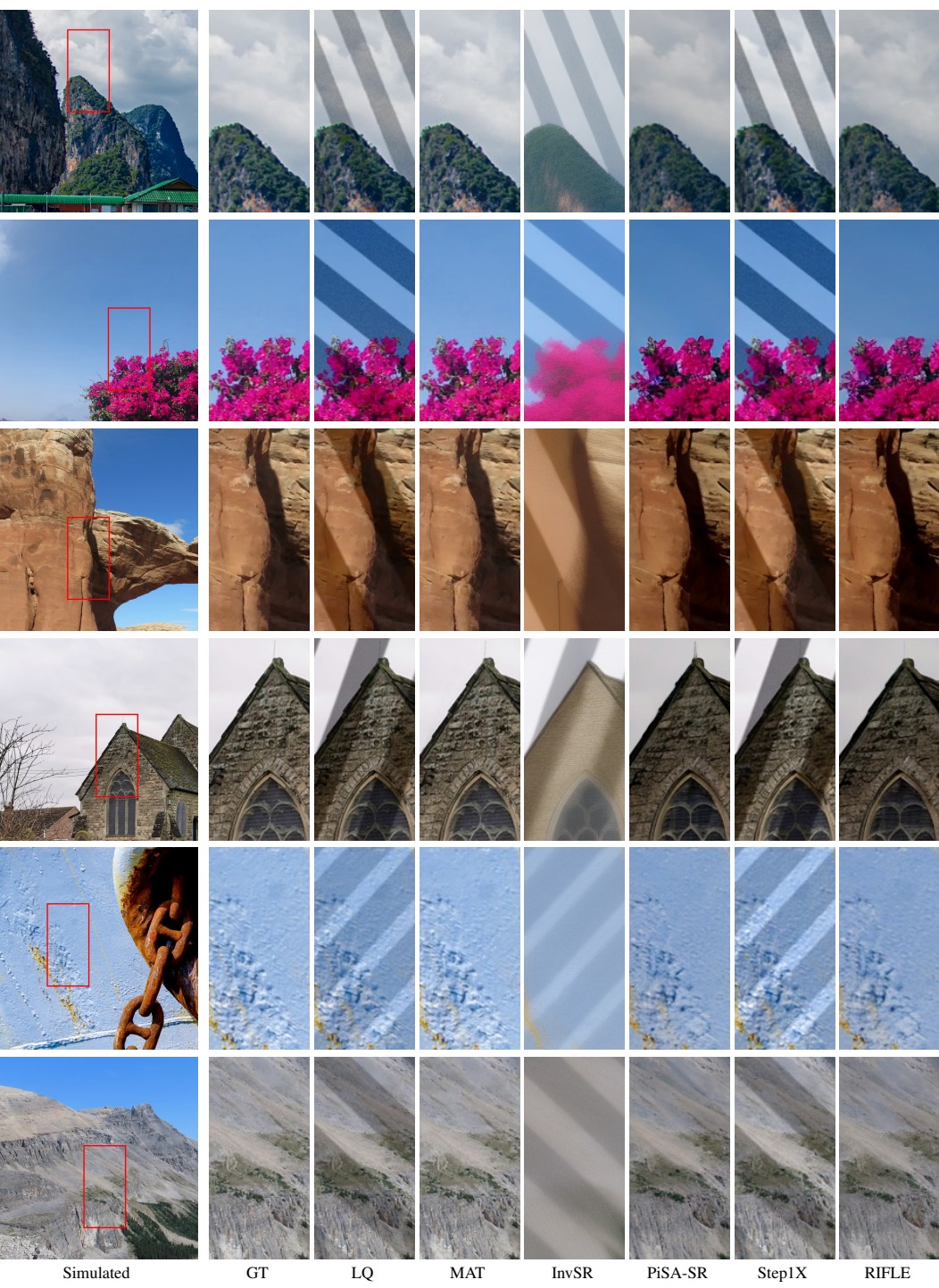

Figure 15: Visual comparison with flicker-banding (LQ), banding-free images (GT), and other debanding methods on our simulated dataset based on LSDIR (Li et al. (2023)).

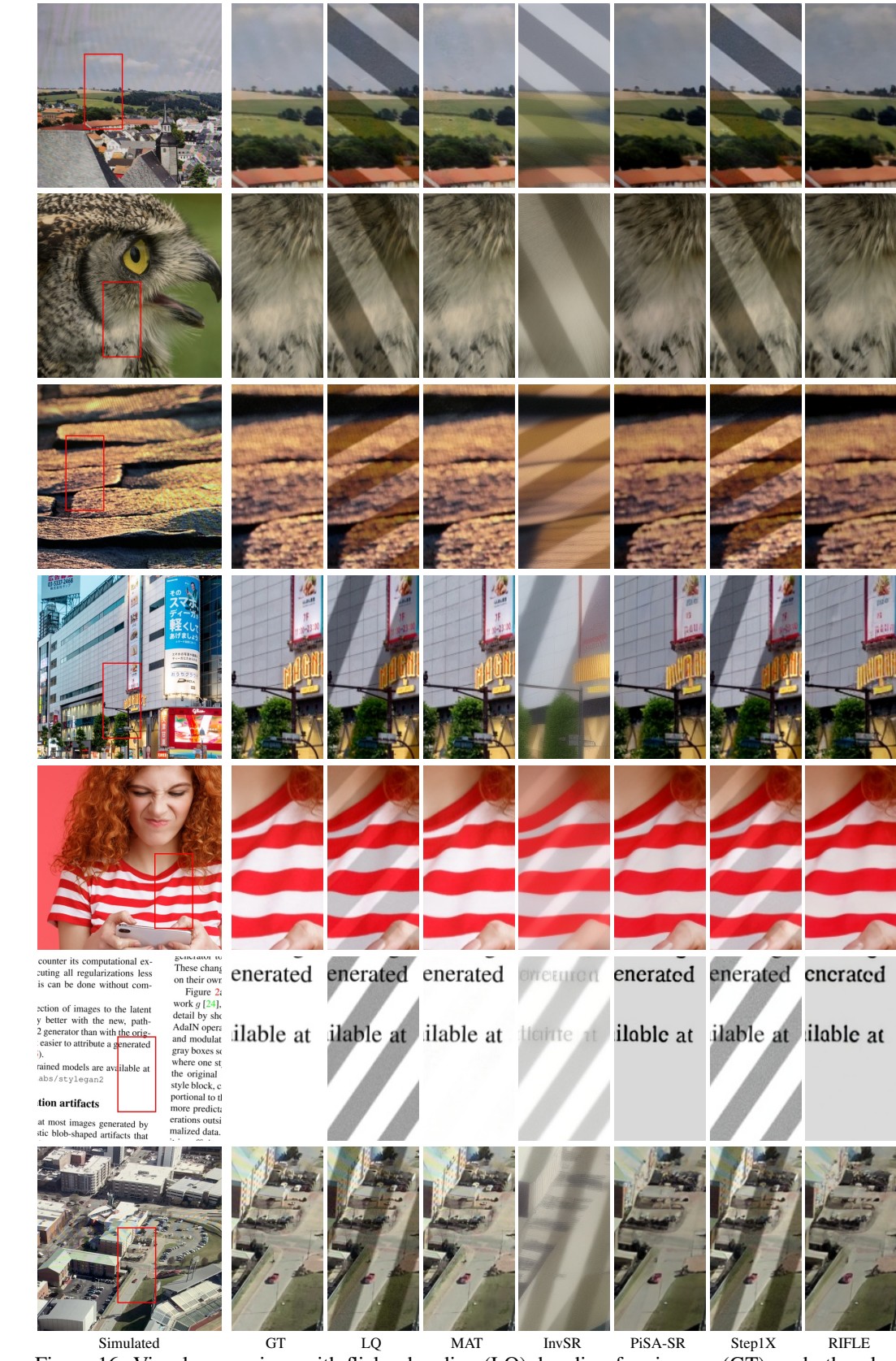

Figure 16: Visual comparison with flicker-banding (LQ), banding-free images (GT), and other de-banding methods on our simulated dataset based on UHDM (Yu et al. (2022)).

# J    LIMITATIONS AND AREAS FOR IMPROVEMENT.

Although our proposed method achieves decent performance in flicker-banding removal, there are still some limitations and areas for improvement:

**Residual Colored Bands.** Some input images may contain extra colored stripes caused by certain display dimming methods. Our model is designed mainly for gray-scale flicker and can leave these colored components visible, as shown in Fig. 17.

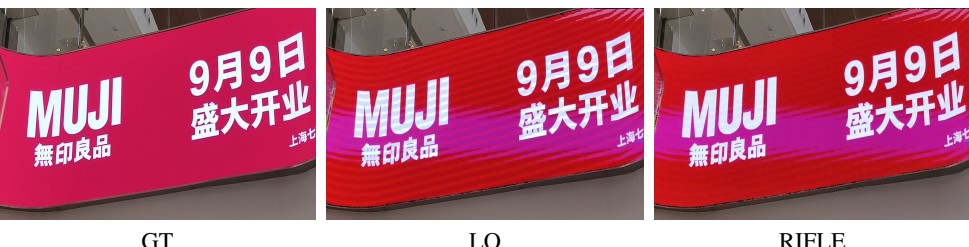

GT        LQ        RIFLE

Figure 17: A case where our method leaves residual magenta colored bands.

This limitation arises from the training simulator, which focuses on luminance modulation. Future work will expand the synthetic dataset to cover a wider range of banding types.

**Texture Detail Loss.** Because our restoration relies on a diffusion process that iteratively refines the image, it can slightly alter original textures or introduce fine details that were not present in the source. As shown in Fig. 18, additional details are generated when removing FB.

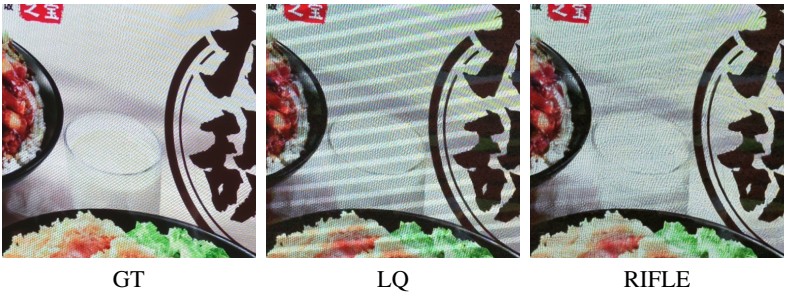

GT        LQ        RIFLE

Figure 18: A case where our method alters fine texture details.

This is a common challenge in diffusion-based restoration methods. Future improvements could be made by combining diffusion with other techniques to protect true scene details.

**Handling Severe or Complex Patterns.** While our method works well on typical flicker-banding patterns, scenes with highly non-stationary, complex, or curved banding may still pose challenges. Purely black banding regions that completely obscure underlying content can also be difficult to recover. In such cases, the model may failed to restore the lost information.

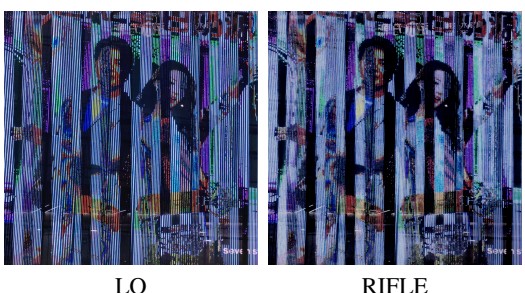

LQ        RIFLE

Figure 19: A case where our method struggles with extremely complex banding patterns.

Improvements could be made by considering multi-frame information, expanding the training data by including more complex banding patterns, or physics-based modeling of display artifacts.

## K OUR SIMULATED AND REAL-WORLD DATASET VISUALIZATION.

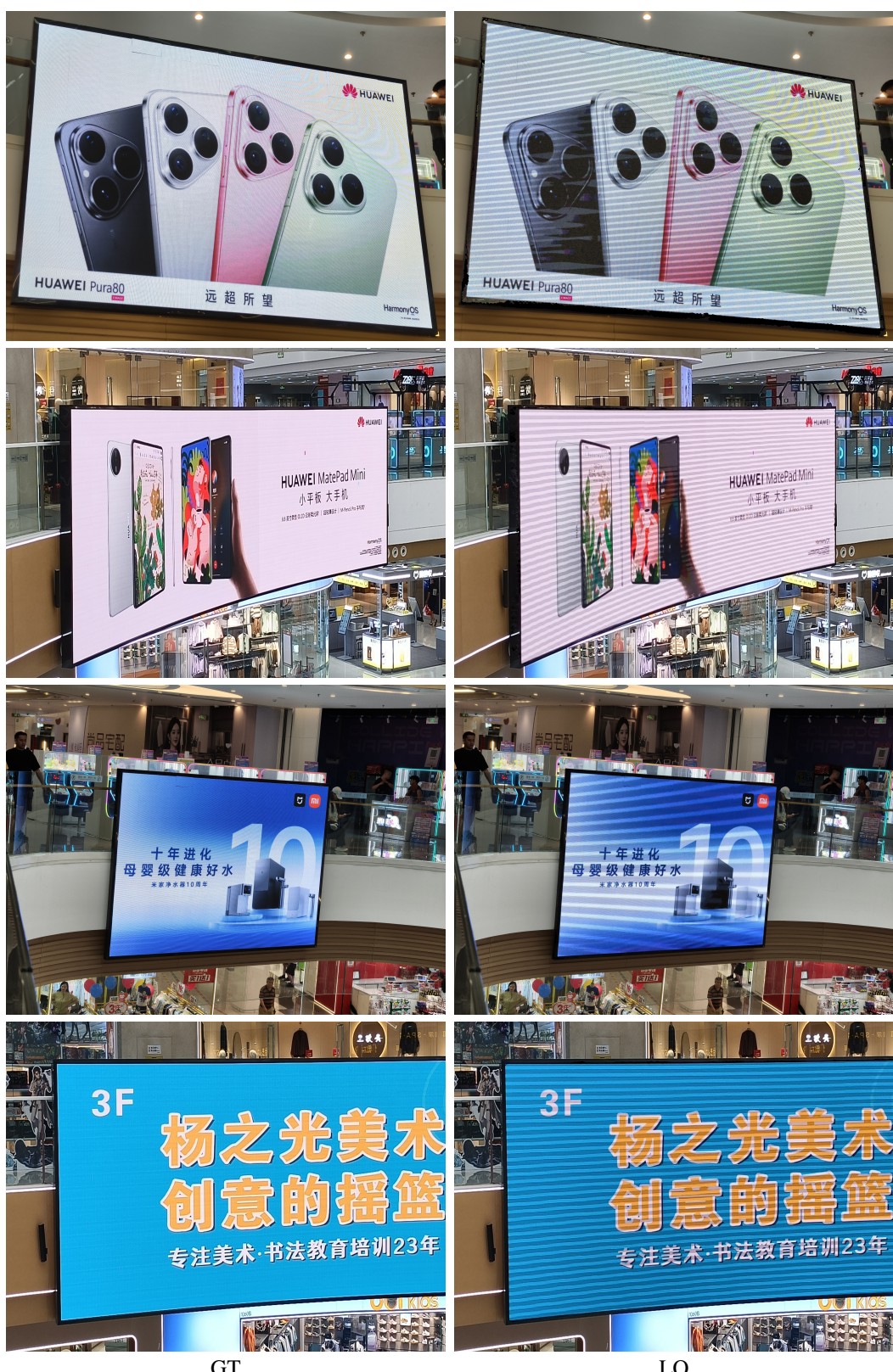

GT        LQ

Figure 20: Our real-world dataset partial visualization.

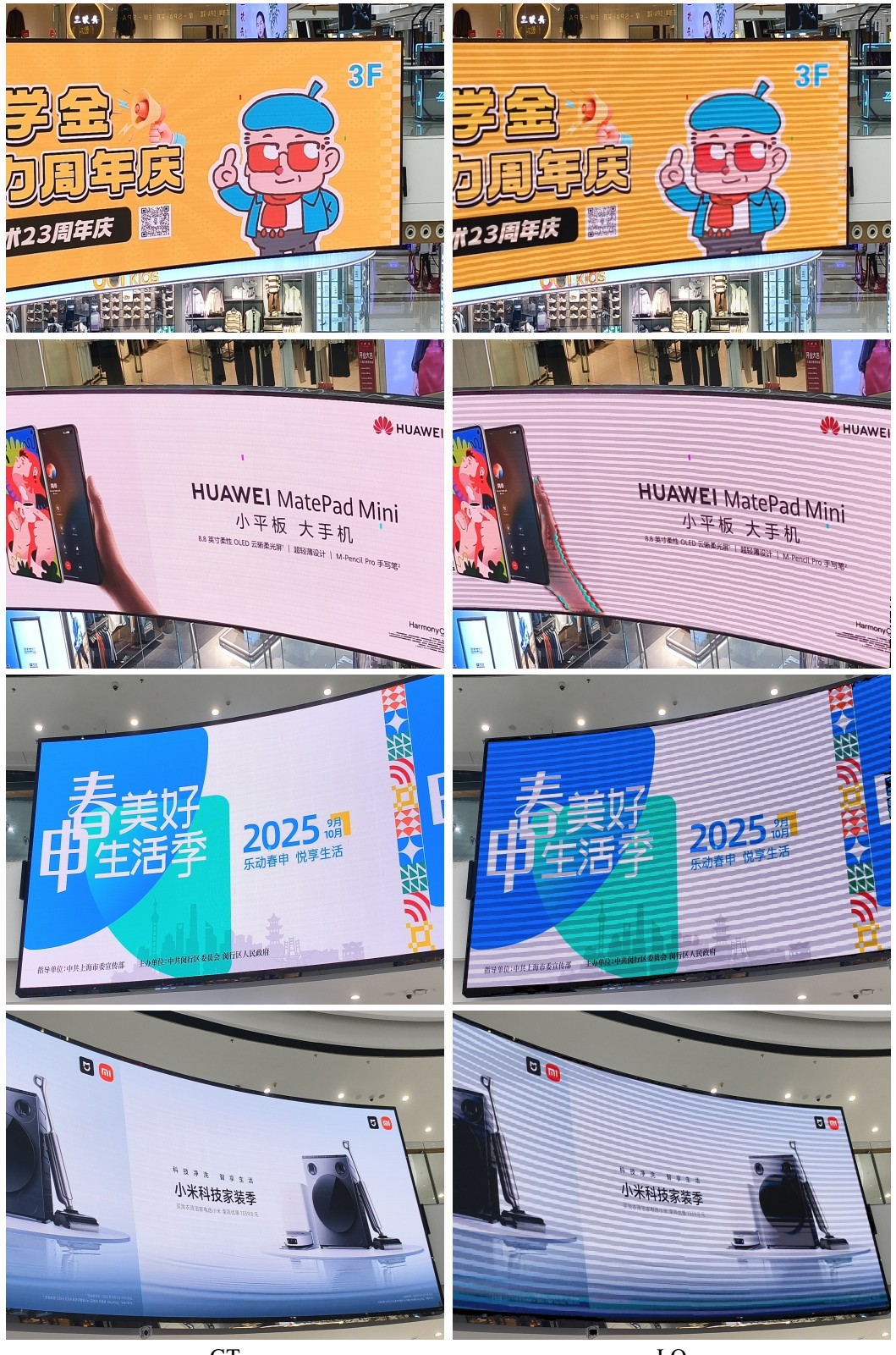

GT                                                      LQ

Figure 21: Our real-world flicker-banding dataset partial visualization.

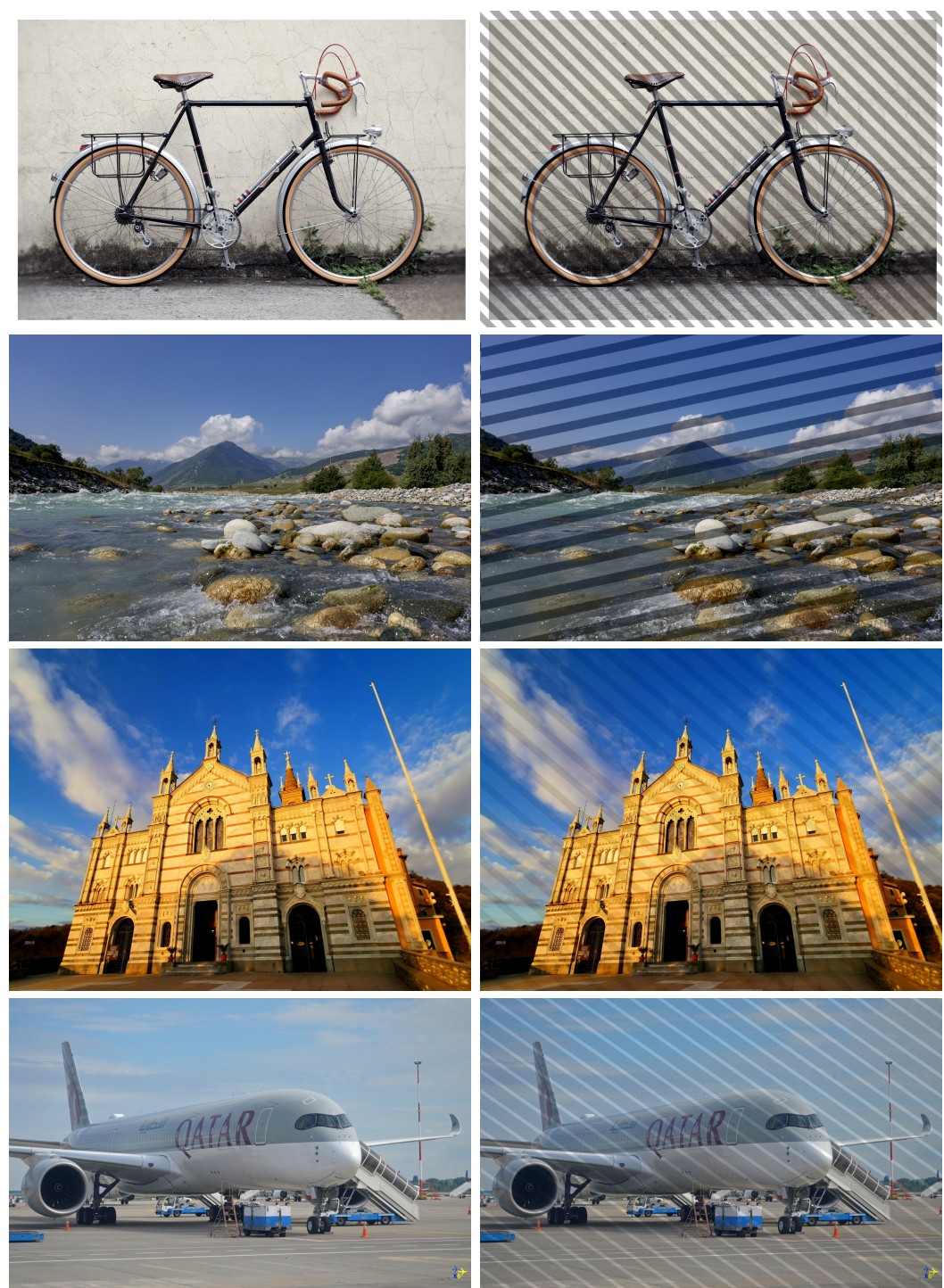

GT                                                  LQ

Figure 22: Our simulated dataset based on LSDIR (Li et al. (2023)) partial visualization.

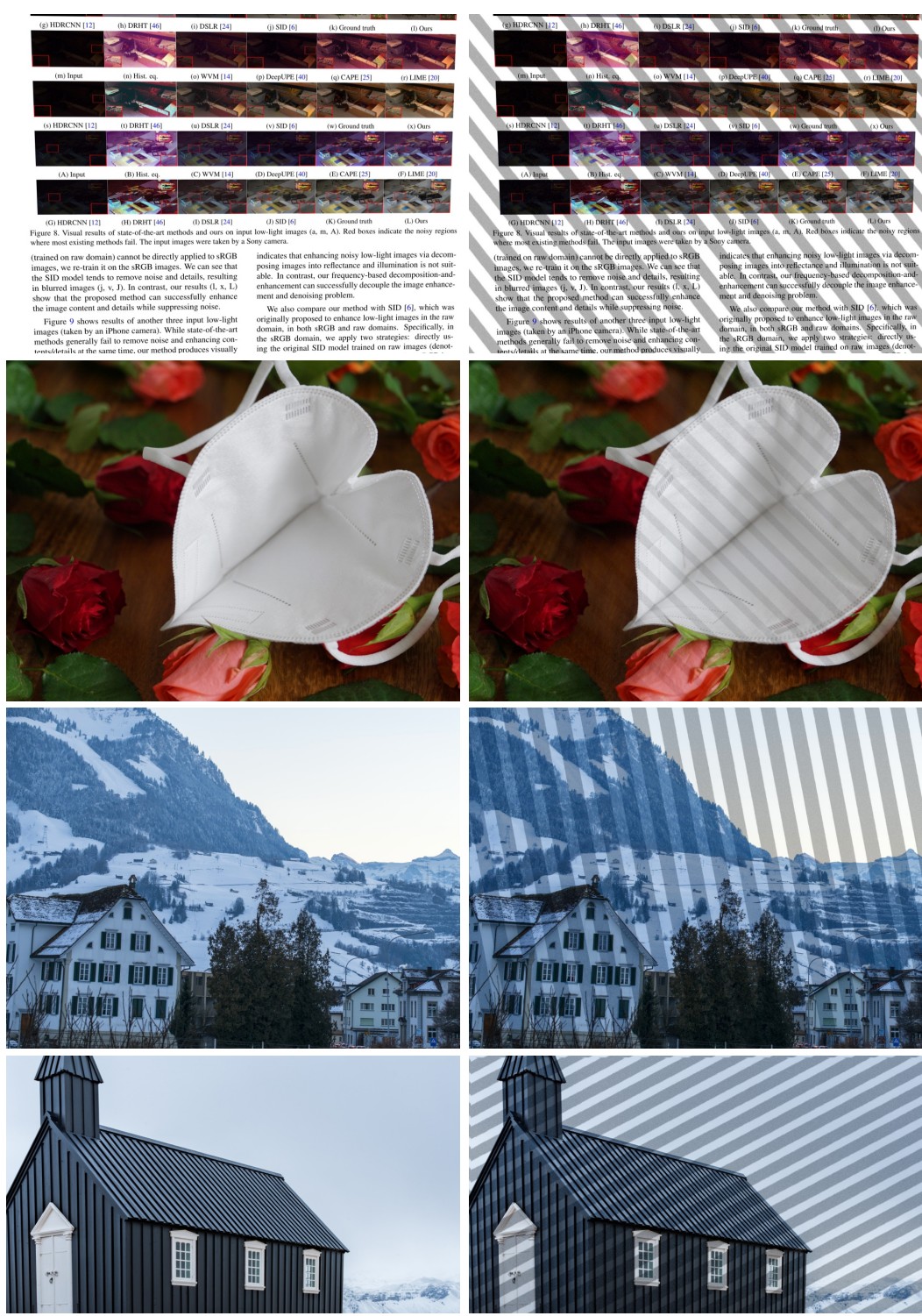

GT

LQ

Figure 23: Our simulated dataset based on UHDM (Yu et al. (2022)) partial visualization.