# OpenReview forum: "RIFLE: Removal of Image Flicker-Banding via Latent Diffusion Enhancement"
_ICLR.cc/2026/Conference — Submitted to ICLR 2026_

### Official Review · Reviewer_XrFB · 2025-10-28

**Soundness:** 3
**Presentation:** 3
**Contribution:** 3
**Rating:** 6
**Confidence:** 2

**Summary:**

The work tackles flicker-banding, alternating bright/dark stripes when photographing emissive displays (OLED/LED/CRT, etc.) with rolling-shutter cameras due to temporal aliasing with PWM/scan refresh.

To overcome the lack of paired data, the authors (i) devise a simulation pipeline that injects FB in the luminance channel with stochastic jitter in angle, spacing, width, feathered boundaries, and sensor noise; and (ii) collect a real-world test set by capturing short-exposure vs. long-exposure pairs from fixed viewpoints, aligned at pixel level.

RIFLE is a one-step latent diffusion restorer (PiSA-SR backbone) augmented with: a Flicker-Banding Prior Estimator (FPE) (ResNet) that predicts band attributes (angle/spacing/width) and injects them into the diffusion network; a Masked Loss (ML) that weights supervision toward banded regions while preserving background fidelity

**Strengths:**

This work tackles a timely and practical problem that is clearly distinct from moiré and common in everyday captures, and it is the first I have seen to set out a coherent task definition, dataset, and baseline specifically for flicker banding. The luminance channel simulation is thoughtfully engineered, with stochastic jitter and feathered transitions, so the artifacts closely match real cases and the pipeline scales well. The approach is task aware: a lightweight prior estimator (FPE) plus a masked loss is a simple and effective recipe, and the ablations show their complementary roles. Results consistently lead on a real world benchmark, visibly reducing banding under both mild and severe conditions while remaining competitive with strong diffusion and transformer baselines. Overall, this is a well rounded and complete paper, and if the authors release the data and code it would likely be helpful to the community.

**Weaknesses:**

1. The method feels like a careful integration of known pieces (one-step diffusion, masked loss, small prior estimator) without a single, clearly novel idea specific to flicker-banding. But that doesn’t mean the paper is without value. I think solving a worthwhile problem with a simple approace and doing it with a complete, well-structured pipelin, is still commendable. So I’d like to see how other reviewers evaluate the problem definition.
2. Assumptions like luminance-only masking and mostly straight stripes may fail with curved or multi-frequency banding, subpixel-dependent chroma effects, or high-nit LED walls. Show out-of-distribution tests and discuss failure modes. Can the entire process now handle more of these related small issues?
3. The real-world benchmark is small and not diverse enough in displays, PWM frequencies, camera models, shutter speeds, and ISO. Add stratified tests and more scenes to show breadth.

**Questions:**

See weaknesses. And I admit that I am not an expert on this topic, so I would like to see other reviewers' opinions on this topic.

---

> ### Author Response · Authors · 2025-11-21
> **Response to Reviewer XrFB**
>
> Q4-1: Simple appoace to solve a worthwhile problem.
> > A4-1: Thank you for your recognition.
> >
> > 1. Considering that we are the first to research the removal of the flicker-banding, constructing a complete pipeline including data collection, simulation process, and model design is more important than delicate idea. We highly praise your following points, "solving a worthwhile problem with a simple approace and doing it with a complete, well-structured pipeline, is still commendable".
> > 2. One of our key contributions is the proposal of our simulation process. The luminance-only assumption makes a big difference to the model performance and largely shortens the gap between the simulated flicker-banding image and real-world one. Our experiments indicate that without the luminance-only assumption, none of the methods including our model, have any effects. Their outputs are close to the inputs, indicating that they can't recognize the real-world flicker-banding and there is a large domain gap between simulated and real-world one without luminance-only assumption. Luminance-only assumption is one of our novel ideas specific to flicker-banding.
> > 3. Flicker-banding parameter estimator ( FPE ) and masked loss ( ML ) also contain our contributions. We localized and implemented the masked loss and small prior estimator, these two ideas in terms of flicker-banding removing models. According to the experimental results of our ablation study, both ML and FPE bring great improvements to the model performance, especially the visual comparison. The effectiveness also indicates that our straightforward idea truly has its own value.
>
> Q4-2: The assumption of simulation process and limitation of benchmark scale and diversity.
> > A4-2: Thanks for pointing out.
> >
> > 1. Our experiments show that in the phone's automatic exposure mode, over 95% of the resulting photos are straight dark stripes. We assume that our real-world dataset truly contains most of flicker-banding categories we encounter in our daily lives and the assumption of simulation process is reasonable.
> > 2. The FB we present in section A of the supplementary material only exists in the extreme scenes ( very short exposure time). They degenerate into a dark straight one as the exposure time increases. Owing to relatively high exposure time we usually use, the dark straight FB is more important to be the research focus. The severe flicker-banding in the extreme scenes will be one of our future research directions.
> > 3. Our real-world dataset contains all kinds of straight dark FB scenes and provides ample coverage for the vast majority of valuable FB scenarios. We have changed the type of displays, camera models, exposure time and ISO to obtain as many kinds of FB as possible. And the luminance-only assumption is also reasonable if we only research on the dark straight flicker-banding.
>
> Q4-3: Out-of-distribution tests.
> > A4-3: Thanks for pointing out.
> >
> > 1. Although straight dark flicker-banding (FB) is our main research topic, we also provide out-of-distribution test results on other severe FB scenes in the section E of the supplementary material.
> > 2. Our method truly suffer great difficulties in deal with these out-of-distribution scenes. On the one hand, the training set of our method doesn't include these severe scenes and the expansion of FB types may bring performance enhancement. On the other hand, many images of these severe scenes lose too much original information, making it impossible to recover the image quality through a single frame.
> > 3. Exploring the solutions to these out-of-distribution cases is also one of our future researching directions. We believe that we can find a multi-frame solution regarding these out-of-distribution severe scenes.

---

> ### Author Response · Authors · 2025-11-26
> **Call for further discussion with Reviewer XrFB**
>
> > Dear Reviewer XrFB,
> >
> > We sincerely appreciate the time you dedicate to reviewing our paper and providing your valuable feedback. We have carefully considered your comments and provided detailed responses:
> > 1. We support your viewpoint, **a simple appoace to solve a worthwhile problem is still commendable**. We also further clarify that **simple and effective** solutions are far more important than intricate and flashy designs for **the first work**.
> > 2. We explain **the assumption of simulation process and limitation of benchmark scale and diversity** further.
> > 3. We provide the results of **the out-of-distribution tests**.
> >
> > As the author–reviewer discussion phase is nearing its end, we would like to confirm whether our responses address your concerns. If any questions or suggestions remain, please feel free to let us know. We would be grateful to hear them.
> >
> > Thank you again for your time and valuable feedback.
> >
> > Best regards,
> >
> > Authors

---

> ### Author Response · Authors · 2025-11-28
> **Second call for discussion with Reviewer XrFB**
>
> > Dear Reviewer XrFB,
> >
> > We sincerely appreciate the time and effort you dedicated to reviewing our work, as well as your insightful comments. We have carefully followed your suggestions, providing additional **out-of-distribution tests** and further explanation on **benchmark diversity**.
> >
> > Due to the approaching rebuttal deadline and Reviewer gzRh responding that our response can address the concerns on the **dataset and simulation process**, we also hope to further discuss with you whether your concerns have been addressed. Please let us know if there are still any parts of our work that remain unclear.
> >
> > Best,
> >
> > Authors

---

### Official Review · Reviewer_gzRh · 2025-10-30

**Soundness:** 2
**Presentation:** 2
**Contribution:** 2
**Rating:** 2
**Confidence:** 4

**Summary:**

This paper introduces RIFLE, a diffusion-based method designed to alleviate flicker-banding (FB) distortions in individual images. These visual anomalies originate from temporal aliasing, a consequence of the interplay between a camera's rolling shutter and the brightness modulation of displays. The authors make three principal contributions: a methodology for creating synthetic FB-corrupted images for training purposes, a fresh, paired real-world dataset for evaluating performance, and the RIFLE model itself. RIFLE is built based on the PiSA-SR baseline by incorporating a Flicker-banding Prior Estimator (FPE) and a Masked Loss (ML) function.

**Strengths:**

* **Valuable Problem:** The paper deals with flicker-banding removal, which is a real and under-explored issue in computational photography. This actually affects everyday users.
* **Novel Dataset:** The authors put together what they say is the first paired real-world dataset for FB removal. That’s a big deal for the community—it gives everyone a solid benchmark to build on.
* **Realistic Simulation Pipeline:** They designed a simulation pipeline that creates realistic FB artifacts by working with the luminance channel and adding random jitter. This is key for training data-driven models.

**Weaknesses:**

The work takes on an important, often overlooked problem and brings a useful new dataset to the table. But RIFLE, the approach the authors introduce, doesn’t really break new ground, and the experiments don’t do a good job proving it works. When you look at the breakdowns, the main piece—FPE—barely makes a difference. Plus, the numbers aren’t shown clearly or consistently, which just makes it harder to figure out how well the method actually performs.

**Major Weaknesses:**
*   **Restricted Methodological Originality and Unproductive Elements:** The RIFLE model is emphasized as a substantial contribution; however, its novelty compared to the PiSA-SR baseline is minimal. The method consists of two enhancements:

    *   **Masked Loss (ML):** This involves a conventional weighted loss function that assigns a greater weight to the areas affected by banding. Considering that the background (non-banded) area is already noticeably similar to the ground truth, this design choice appears straightforward and simple, rather than a significant innovation.
    *   **Flicker-banding Prior Estimator (FPE):** This is the main component, a ResNet-15 architecture trained to predict three banding attributes (width, spacing, angle). However, the paper's component analysis shows the near-ineffectiveness of this element. The baseline (PiSA-SR) achieves a PSNR of 22.12. The addition of the FPE yields a minor improvement of 0.03 dB (PSNR of 22.15). Conversely, incorporating the simple ML alone results in a PSNR of 22.28. The complete RIFLE model (ML+FPE) attains a PSNR of 22.30, suggesting that the entire 0.18 dB improvement over the baseline is due to the uncomplicated masked loss, rather than the prior estimator. This finding directly contradicts the paper's central technical claim.

*   **Contradictory and Vague Quantitative Results:** A notable inconsistency exists between the numerical results presented in Table 1 and Table 2.

    *   Table 1 presents results on the "real-world flickering-banding datasets." Here, RIFLE achieves a PSNR of 20.66, whereas the PiSA-SR baseline achieves a PSNR of 20.57.
    *   Table 2 presents the ablation study, which, according to Figure 7's caption, is also performed "on our Real-Flicker dataset." Here, RIFLE (as "ML+FPE") achieves a PSNR of 22.30, and the baseline (PiSA-SR) achieves a PSNR of 22.12.

    These two tables, supposedly reporting on the same dataset, exhibit a significant and unexplained difference of approximately 1.6-1.7 dB for the same models. It is unclear whether Table 2 uses a smaller subset of the data or if a more profound methodological mistake exists. This discrepancy makes it impossible to assess the model's true capabilities or the validity of the component analysis.

*   **Inherently Limited Method Design:** The simulation pipeline's crucial observation involves applying banding exclusively to the luminance (Y) channel, based on analysis in Appendix C. While clever, this design choice introduces a critical constraint. The authors acknowledge in their limitations (Appendix E) that the model is primarily intended for "gray-scale flicker" and struggles with colored stripes. This is a major disadvantage, as the paper's own survey in Appendix A acknowledges that technologies like OLEDs and projectors do generate chromatic artifacts.

    Furthermore, the FPE is designed to predict a single banding angle. This design lacks the ability to handle the complex or curved banding patterns prevalent in real-world captures, a limitation also conceded by the authors. As a result, the method is ill-suited for many real-world instances of the very problem it claims to solve.
* Weak Ablation Studies and Comparison: no runtime, model complexity comparison.

**Minor Weaknesses:**
*   **Exaggerated "First Work" Assertion:** The paper repeatedly claims to be the "first work" on FB simulation and removal. This claim is debatable. While it may be the first to coin the term "Flicker-Banding" and leverage a diffusion model, the underlying phenomenon is a known artifact resulting from rolling shutters interacting with modulated light sources. The related work section is brief and quickly dismisses comparable work, such as single-image deflicker techniques, which also attempt to eliminate flicker-induced spatial distortions from a single frame.

**Questions:**

* Can authors explain why do Table 1 and Table 2 look so different even both claim to use the real-world dataset? As mentioned above, the PSNR numbers in Table 2 are higher by about 1.6 or 1.7 dB.
* Aslo, as stated above, FPE is reported to improve performance by 0.03dB, while Masked Loss gives bigger jump of 0.16dB. Can you explain why FPE is important for RIFLE model if its effect is so small?
* Can authors provide more comparison of runtime complexity, I doubt that this will consume a lot of time, which is not really phones friendly, which contradict with the original purpose of removing flickers when capturing by phone.

---

> ### Author Response · Authors · 2025-11-21
> **Response to Reviewer gzRh (Part 1)**
>
> Q3-1: Restricted Methodological Originality and Unproductive Elements.
> > A3-1: Thanks for your question.
> > 1. We are the first to raise the flicker-banding problem in the academia. As for a barnd-new topic, it is more important to construct an effective pipeline to solve the problem rather than have particularly ingenious ideas in terms of technical details. A completely new process, including dataset construction, simulation and model design, is itself a major innovation. Reviewer XrFB also agrees that "solving a worthwhile problem with a simple approace and doing it with a complete, well-structured pipeline, is still commendable".
> > 2. As for the removing model, both of Masked Loss (ML) and Flicker-banding Prior Estimator (FPE) are specially designed for the removal of flicker-banding and experiments have proven their effectiveness. Regarding the experiments results, for a low-level vison issue, the visual comparison indicates the true performance of the model and is much more important than the quantative results. We also mention that none of the evaluation metrics can reflect the degree of flicker-banding precisely and analysis only on the quantitative results, neglecting the visual comparison, is meaningless. In Fig.6 in the main paper, RIFLE gains a great improvement over PiSA-SR. There are still obvious stripes remaining in the results of PiSA-SR, and the results of RIFLE remain clean and uniform. It indicates the introduction of our ML and FPE truly makes a difference, which stands for our contribution.
> > 3. Similarly, analyzing each component's effect only through the quantitative results in the ablation study is unreasonable. The evaluation metrics only reflect the distance between the recovered image and banding-free image, rather than the degree of flicker-banding. So we also provide the visual comparison of the ablation study. For the first line, the introduction of FPE makes a great difference to the baseline, making the stripe much lighter in color. For the second line, the introduction of FPE brings great improvements to the ML, making the stripes almost totally removed. Therefore, our ablation study results can reflect our ML and FPE's effectiveness.
>
> Q3-2: Contradictory and Vague Quantitative Results.
> > A3-2: Thanks for pointing out. Ther are many understandings.
> >
> > 1. As mentioned in the section 4.1.2, the real-world flicker-banding dataset has full-sized and cropped version. The main experiments in Table 1 are based on the cropped version of our real-world dataset, and the ablation study experiments in Table 2 are based on the full-size version of our real-world dataset. Comparing these two tables directly is unreasonable because the evaluation metrics of model input (LQ) are different.
> > 2. To avoid the misunderstanding, we add the dataset version (cropped or full-size) to the headings of these two tables. We guarantee that the full real-world dataset is used throughout the testing process.
> > 3. If you persist in the comparison of the Table 1 and Table 2, we can also provide the main results on full-size version of real-world dataset.
> >
> >     | Methods | PSNR $\uparrow$  | SSIM $\uparrow$ | ms-SSIM $\uparrow$ | LPIPS $\downarrow$ | DISTS $\downarrow$ | FSIM $\uparrow$ | GMSD $\downarrow$ |
> >     |:--------:|:--------:|:--------:|:--------:|:--------:|:--------:|:--------:|:--------:|
> >     | LQ | 21.78 |  0.7373  | 0.7655 | 0.2322 | 0.1219 | 0.8594 | 0.1848 |
> >     | MAT | 22.25 |  0.7480 |  0.8124 | 0.2104 | 0.1112 | 0.8832 | 0.1590 |
> >     | InvSR | 19.18 | 0.6254 | 0.7243 | 0.3805 | 0.1811 | 0.8358 | 0.2089 |
> >     | Step1X | 21.15 | 0.6951 | 0.7549 | 0.2975 | 0.1307 | 0.8563 | 0.1880 |
> >     | PiSA-SR| 22.12 | 0.7490 | 0.8677 | 0.1812 | 0.0920 | 0.9406 | 0.1297|
> >     | RIFLE (ours) | 22.30 | 0.7425 | 0.8697 | 0.1902 | 0.0908 | 0.9460 | 0.1286|
> >
> > 4. We have to reaffirm the validity of the results of our experiment. They can prove our model's true capabilities and the validity of the component analysis.

---

> ### Author Response · Authors · 2025-11-21
> **Response to Reviewer gzRh (Part 2)**
>
> Q3-3: Explanation on simulation pipeline and banding types.
> > A3-3: Thanks for pointing out.
> >
> > 1. The FB we present in the section A of the supplementary material only exists in the extreme scenes (very short exposure time). They degenerate into dark straight one as the exposure time increases. Owing to relatively high exposure time we usually use, the dark straight FB is more important to be the research focus. So the luminance-only assumption in the simulation process is reasonable.
> > 2. The severe flicker-banding in the extreme scenes will be one of our future research directions, but not this work. It is also impossible to remove these severe FB from a single frame. We are searching for a multi-frame solution.
> > 3. Our simulation pipeline and real-world dataset contain all kinds of straight dark FB scenes, and provide ample coverage for the vast majority of valuable FB scenarios.
>
> Q3-4: Weak Ablation Studies and Comparison.
> > A3-4: Thanks for your suggestion.
> > 1. We provide the runtime and model complexity comparison of PiSA-SR and RIFLE in the following table. The runtime results are obtained with an input image sized 512 $\times$ 512 on the same device.
> >
> >     | Model | Runtime (s) | Params (M) |
> >     |:--------:|:--------:|:--------:|
> >     |PiSA-SR | 0.1124 | 949.6 |
> >     | RIFLE | 0.1188 | 995.6 |
> > 2. The increase of runtime and params brought by the introduction of FPE is relatively low, indicating that our proposed component is light-designed, costing little extra expense.
> > 3. The 0.1188-second inference time is sufficient for efficiently processing large batches of images.
> > 4. Our method is to remove the flicker-banding on the photos that have already been taken, acting as a post-processing algorithm, rather than removing flickers when capturing by phone. It can be deployed to the cloud and does not have high requirements for real-time performance.
> >
> Q3-5: First Work Assertion.
> > A3-5:
> > Thanks for the comment. While terminology can overlap, to avoid any ambiguity, we clarify that although some prior works in *Section 2* contain terms such as *“deflicker”* or *“destriping”*, these methods address fundamentally different degradations.
> >
> > - Existing **single-image deflicker** methods focus on global lighting fluctuations caused by AC-powered lamps, which are horizontal, quasi-periodic sinusoidal variations across the entire image.
> > - Similarly, **destriping/debanding** techniques handle sensor-originated fixed-pattern noise or quantization artifacts, which are typically static and do not involve temporal modulation.
> > - Other **rolling-shutter restoration** works address geometric distortions due to camera and object motion during capture, which do not result in the striped artifacts characteristic of FB.
> >
> > In summary, prior works do not model the display-driver PWM/scan × rolling-shutter temporal–spatial aliasing at the core of the FB phenomenon, nor do they provide task-specific definitions, simulation, or paired real data. Our work is the first to systematically formulate this degradation and build a complete benchmark and solution tailored to it.

---

> > ### Comment · Reviewer_gzRh · 2025-11-24
> >
> > I appreciate the authors' thorough explanation of the difference between Tables 1 and 2 (Question 3-2).
> >
> > 1. Regarding the Model Effectiveness and Quantitative Discrepancy (A3-2 \\& A3-1)
> > Thank you for clarifying that full-sized data was used in Table 2 and cropped data was used in Table 1. This distinction explains a lot. However, the rebuttal's full-size results have actually confirmed my main concern about the Flicker-banding Prior Estimator (FPE) technical contribution.
> >
> > Using the full-size dataset figures from the paper's Table 2 and rebuttal:
> >
> > - Masked Loss (ML) + Baseline: 22.28 PSNR (Table 2)
> > - RIFFLE (ML + FPE + Baseline): 22.30 PSNR (Rebuttal Table)
> >
> > The FPE module's quantitative contribution is only **0.02 dB**. This is likely within the margin of training variance or random seeds and is statistically insignificant.
> >
> > Although the authors contend in A3-1 that "visual comparison is much more important", claiming that a module is "effective" and "specially designed" when it produces essentially no quantitative improvement over a basic weighted loss function seems scientifically unreasonable. The LPIPS or SSIM scores should show improvements if the FPE were genuinely removing artifacts that the Masked Loss overlooked, but these improvements are likewise negligible (e.g., LPIPS improves by only 0.0053).
> >
> > 2. Regarding Originality in Methodology (A3-1)
> > For a new problem, the authors argue that "a simple pipeline" is sufficient. I agree that the simulation pipeline and dataset are contributions. However, RIFLE is presented in the paper as a unique methodological contribution. Based on the data, RIFLE is essentially PiSA-SR + weighted loss. The FPE is an architectural addition that, without measurable gain, increases inference time and parameter count.
> >
> > 3. Regarding Simulation and Scope (A3-3)
> > I agree with the authors that straight/grayscale banding is common. The simulation pipeline’s restriction to luminance-only artifacts is still a disadvantage for a general "flicker-banding" solution, though I accept it as a reasonable scope for a first work, especially since Appendix A specifically highlights colored artifacts in OLEDs.
> >
> > Conclusion:
> > The problems with data presentation have been clarified by the rebuttal, which is appreciated. However, the clarified data demonstrates the inefficiency of the proposed technical novelty (FPE). The dataset and simulation pipeline constitute the majority of the paper’s value. The claims made in this methods paper about the RIFLE architecture are not supported by the results presented. Although I will consider the dataset contribution positively, my concerns about the validity of the method’s design still remain.

---

> ### Author Response · Authors · 2025-11-25
> **Further discussion with  Reviewer gzRh**
>
> Q3-6: Clarification on the quantitative contribution.
> >A3-6: Thanks for your reply to our rebuttal.
> >1. We add clarification on the quantitative results to section B in the supplementary material. It indicates the saturation of the evaluation metrics for the FB. Even with a great performance in visual comparison as we present, there is still a minor improvement in quantitative results. Therefore, we believe that the minor improvement in quantitative results brought by FPE can't indicate its inefficiency, considering its great contribution to the visual comparison.
> >2. We also provide the results of the user study, which shows that our model's performance is significantly improved after introducing a small number of parameters and time overhead, differing from the numerical comparison. This can also be understood as another aspect of the numerical results.

---

> ### Author Response · Authors · 2025-11-28
> **Call for further discussion with Reviewer gzRh**
>
> > Dear Reviewer gzRh,
> >
> > We sincerely appreciate the time and effort you dedicated to reviewing our work, as well as your insightful comments. Following your suggestions, we have provided additional clarification on the **quantitative results** and the **effects of our FPE**.
> >
> > As the rebuttal deadline is approaching, we would also like to check with you whether our responses have addressed your concerns. If there are still any parts of our work that remain unclear, please feel free to let us know.
> >
> > Best,
> > Authors

---

### Official Review · Reviewer_tUpL · 2025-11-01

**Soundness:** 3
**Presentation:** 3
**Contribution:** 2
**Rating:** 4
**Confidence:** 4

**Summary:**

This paper investigates flicker-banding artifacts (FB) in screen photography, a type of image degradation. This degradation, characterized by bright and dark stripes, arises from the temporal aliasing between display brightness modulation and the camera's rolling shutter mechanism. To address this problem, the paper proposes an FB simulation data generation pipeline and a latent diffusion-based image restoration framework RIFLE.

**Strengths:**

The paper proposes a reasonable data generation pipeline that successfully generated a corresponding dataset. Furthermore, an effective method for removing FB was introduced.

**Weaknesses:**

1. Regarding the dataset, an analysis would be beneficial. Given that 400+ paired patches constitute a relatively small scale, does this dataset adequately cover all possible scenarios where flicker-banding might occur?
2. Currently, all the visualization cases you've presented are screen captures. Screen captures are typically affected by a complex combination of moiré patterns and flicker artifacts. While current moiré removal methods aim to directly produce clean, high-quality images. However, RIFLE, based on your description, only removes flicker artifacts. Consequently, the visualization results still appear to be affected by moiré patterns. This raises the question: Does this imply that your method is a 'degraded version' of moiré removal methods? Furthermore, for an image degraded by screen capture, wouldn't applying a moiré removal method be more appropriate? You lack examples demonstrating scenarios where only flicker artifacts are present without moiré interference. Simply showcasing screen-capture scenarios is insufficient to fully demonstrate the necessity and unique value of your method.

**Questions:**

Please refer to Weaknesses.

---

> ### Author Response · Authors · 2025-11-21
> **Response to Reviewer tUpL**
>
> Q2-1: The limitation of benchmark scale and diversity.
> > A2-1: Thanks for pointing out.
> > 1. Our experiments show that in the phone's automatic exposure mode, over 95% of the resulting photos are straight dark stripes. We assume that our real-world dataset truly contains most of flicker-banding categories we encounter in our daily lives.
> > 2. The FB we present in section A of the supplementary material only exists in the extreme scenes ( very short exposure time). They degenerate into a dark straight one as the exposure time increases. Owing to relatively high exposure time we usually use, the dark straight FB is more important to be the research focus. The severe flicker-banding in the extreme scenes will be one of our future research directions.
> > 3. Our real-world dataset contains all kinds of straight dark FB scenes and provides ample coverage for the vast majority of valuable FB scenarios. We have changed the type of displays, camera models, exposure time, and ISO to obtain as many kinds of FB as possible.
>
> Q2-2: The difference between flicker-banding and moiré.
> > A2-2: Thanks.
> > 1. Flicker-banding and moiré patterns are two independent phenomena.
> >
> >     | Aspect    | Flicker-banding (FB)                                                                 | Moiré patterns                                                                                 |
> >     |-----------|-----------------------|--------------------------------|
> >     | Cause     | Caused by **rolling shutter** capturing the **PWM/scanning** of the screen under short exposure | Caused by **interference  between two grids/patterns** with slightly different frequencies or angles |
> >     | Visual form | Appears as **regular alternating light and dark stripes**                          | Appears as **irregular wave-like patterns**, often with **color differences**                 |
> >     | Scenario  | Mainly occurs when **shooting a brightness modulated screen**             | Can appear in **most scenes involving fine repetitive patterns** |
> >     | Signal formation | **Spatio-temporal artifact** | **Purely spatial aliasing** |
> >     | Solution | Little related works | Already many mature demoiré models |
> >
> > 2. We admit that certain screen captures in our dataset are typically affected by a complex combination of moiré patterns and flicker artifacts. This is because moiré patterns are widely present in photos taken from a screen, which makes moiré inevitable during our dataset collection process. And our research target is to remove flicker-banding, not moiré. So it is reasonable that there are severe moirés in our processed results.
> > 3. We also provide the experimental results of demoiré methods. ESDNet is an outstanding demoiréing method accepted by ECCV 2022. It is obvious that demoiré methods don't demonstrate good performance. They fall far behind our method in terms of numerical metrics. For visual comparison, their results still have heavy stripes, while our method leaves almost no stripes. It indicates that our method is not a 'degraded version' of moiré removal methods but an independent method for removing FB.
> >
> >     | Methods | PSNR $\uparrow$  | SSIM $\uparrow$ | ms-SSIM $\uparrow$ | LPIPS $\downarrow$ | DISTS $\downarrow$ | FSIM $\uparrow$ | GMSD $\downarrow$ |
> >     |:--------:|:--------:|:--------:|:--------:|:--------:|:--------:|:--------:|:--------:|
> >     | LQ | 19.43 | 0.5636 | 0.6364 | 0.3374 | 0.2213 | 0.7907 | 0.2091 |
> >     |ESDNet  | 20.49 | 0.6077 | 0.7472 | 0.2782 | 0.2075 | 0.8424 | 0.1587|
> >     | RIFLE (ours) | 20.66 | 0.6220 | 0.8067 | 0.2456 | 0.1723 | 0.8711 | 0.1433 |
> >
> > 4. We believe that removing moiré and flicker-banding simultaneously is an even more challenging task. The superposition of two independent degradation modes cannot be removed by the linear superposition of their removal models. And removing moiré and flicker-banding simultaneously is also one of our future research directions.
>
> Q2-3: The lack of only flicker artifacts scenarios.
> > A2-3: Thanks for your advice.
> > 1. As mentioned in A2-2, moiré patterns are inevitable during capturing the screens. So the appearance of flickering in photographs is accompanied by moiré patterns, and it is difficult to separate them.
> > 2. In our results, there are many patches where the degree of moiré pattern is low and the degree of FB is high. They stand for scenes with only FB artifacts without moiré in some way.
> > 3. We provide the experiment results of the demoiré model performance comparison in the case of Flicker-banding. The existence of flicker-banding (FB) is a great barrier for the demoiré model to finish the task of removing moiré patterns. This suggests that finding a method to remove flicker-banding (FB) is essential, as existing approaches, such as demoiré models, not only fail to address FB scenes but also encounter significant challenges in performing their intended tasks.

---

> ### Author Response · Authors · 2025-11-26
> **Call for further discussion with Reviewer tUpL**
>
> > Dear Reviewer tUpL,
> >
> > We sincerely appreciate the time you dedicate to reviewing our paper and providing your valuable feedback. We have carefully considered your comments and provided detailed responses:
> > 1. We provide explanation on **the scale and diversity of our dataset**.
> > 2. We explain **the difference between the flicker-banding and moire** further.
> > 3. We address your concern on **the lack of only flicker artifacts scenarios**.
> >
> > As the author–reviewer discussion phase is nearing its end, we would like to confirm whether our responses address your concerns. If any questions or suggestions remain, please feel free to let us know. We would be grateful to hear them.
> >
> > Thank you again for your time and valuable feedback.
> >
> > Best regards,
> >
> > Authors

---

> ### Author Response · Authors · 2025-11-28
> **Second call for discussion with Reviewer tUpL**
>
> > Dear Reviewer tUpL,
> >
> > We sincerely appreciate the time and effort you devoted to reviewing our work, as well as your thoughtful comments. Based on your suggestions, we have carried out additional experiments to further clarify **the distinction between flicker-banding and moiré**.
> >
> > As the rebuttal deadline is approaching, and Reviewer gzRh has indicated that our response adequately addresses concerns regarding the **dataset and simulation pipeline**, we would also like to check with you whether your concerns have been fully resolved. If there are still any aspects of our work that remain unclear, please feel free to let us know.
> >
> > Best,
> > Authors

---

### Official Review · Reviewer_qfME · 2025-11-01

**Soundness:** 3
**Presentation:** 2
**Contribution:** 2
**Rating:** 4
**Confidence:** 2

**Summary:**

The paper proposes RIFLE (Removal of Image Flicker-Banding via Latent Diffusion Enhancement), a diffusion-based method to remove flicker-banding (FB)—alternating bright-dark stripes caused by temporal aliasing between camera rolling-shutter and screen brightness modulation. Unlike moiré patterns, FB remains underexplored despite its significant impact on image readability. To address the lack of datasets, the authors create a realistic simulation pipeline to synthesize FB artifacts and construct a paired real-world dataset for rigorous evaluation. RIFLE introduces two novel components: a Flicker-Banding Prior Estimator (FPE) that predicts banding parameters, and a Masked Loss (ML) targeting restoration in banded regions. Experimental results demonstrate that RIFLE outperforms recent image reconstruction baselines in both quantitative metrics and visual quality on real-world FB datasets, making it the first comprehensive attempt at tackling FB using neural networks

**Strengths:**

- Problem framing. Clearly formulates FB removal as distinct from moiré and global deflicker, with a concise physical explanation and taxonomy of patterns.
- Real‑world benchmark: paired captures with short/long exposure and pixel‑level alignment; evaluation limited to screen regions to avoid metric saturation
- Well‑motivated masked loss focusing on banded areas and a lightweight prior estimator (FPE) that predicts angle/spacing/width and conditions the diffusion network.

**Weaknesses:**

- Benchmark scale and diversity. The real test set is small and limited to static screens; this constrains claims about generalization across displays (PWM frequencies/duty cycles/scan schemes), camera models, and capture settings.
- Baselines. Comparisons are to general reconstruction/editing methods (MAT, InvSR, PiSA‑SR, Step1X). Stronger task‑proximity baselines are missing: (i) simple frequency‑domain notch filtering aligned to the estimated banding angle; (ii) line‑pattern destriping/deraining networks; (iii) rolling‑shutter correction plus tone‑mapping heuristics.
- Evaluation scope. Since FB impairs readability, an OCR‑based metric (accuracy on cropped text regions) or a user study on legibility would be very persuasive. The current reference metrics can be insensitive when the non‑banded regions dominate (the paper notes saturation; Sec. 5.2), yet no downstream/readability metric is provided.

**Questions:**

Simulation assumptions. Banding is applied only to luminance, with justification deferred to the supplement. Some PWM/scan schemes can induce color‑dependent artifacts (sub‑field driving). Please quantify the luminance‑only assumption on real data, and report a domain‑gap analysis between simulated and real FB.

---

> ### Author Response · Authors · 2025-11-21
> **Response to Reviewer qfME**
>
> Q1-1: The limitation of benchmark scale and diversity.
> > A1-1: Thanks for pointing out.
> >
> > 1. Our experiments show that in the phone's automatic exposure mode, over 95% of the resulting photos are straight dark stripes. We assume that our real-world dataset truly contains most of flicker-banding categories we encounter in our daily lives.
> > 2. The FB we present in  section A of the supplementary material only exists in the extreme scenes (very short exposure time). They degenerate into dark straight one as the exposure time increases. Owing to relatively high exposure time we usually use, the dark straight FB is more important to be the research focus. The severe flicker-banding in the extreme scenes will be one of our future research directions.
> > 3. Our real-world dataset contains all kinds of straight dark FB scenes and provides ample coverage for the vast majority of valuable FB scenarios. We have changed the type of displays, camera models, exposure time, and ISO to obtain as many kinds of FB as possible.
>
> Q1-2: Stronger task‑proximity baselines comparison.
> > A1-2: Thanks for your suggestion.
> >
> > | Methods | PSNR $\uparrow$  | SSIM $\uparrow$ | ms-SSIM $\uparrow$ | LPIPS $\downarrow$ | DISTS $\downarrow$ | FSIM $\uparrow$ | GMSD $\downarrow$ |
> > |:--------:|:--------:|:--------:|:--------:|:--------:|:--------:|:--------:|:--------:|
> > | LQ | 19.43 | 0.5636 | 0.6364 | 0.3374 | 0.2213 | 0.7907 | 0.2091 |
> > | Notch Filter | 17.86 | 0.4067 | 0.5527 | 0.5340 | 0.3101 | 0.6734 | 0.2172 |
> > | NeRD-Rain | 20.30 | 0.6066 | 0.7447 | 0.2830 | 0.1950 | 0.8462 | 0.1702 |
> > | RIFLE (ours) | 20.66 | 0.6220 | 0.8067 | 0.2456 | 0.1723 | 0.8711 | 0.1433 |
> >
> > 1. We have added more task‑proximity methods' experimental results to the supplementary results, and none of them are a great approach to remove flicker-banding.
> > 2. As for the notch filter, it doesn't achieve any substantial effect and instead damages the image quality. Not only is the original image information damaged, but the minor prediction error of FPE may lead to great residues.
> > 3. As for the destriping/deraining networks, we choose the NeRD-rain, a CVPR 2024 deraining work, as a representative. Although the disordered fine rain streaks are similar to the flicker-banding in some way, the deraining network doesn't work as well as we expected.
> > 4. We assume that the rolling‑shutter correction plus tone‑mapping heuristics already exist in our mobile phone' ISP, and occurrence of numerous flickering phenomena in our daily photography experience indicates its ineffectiveness.
>
> Q1-3: Additional Evaluation Scope.
> > A1-3: Thanks for pointing out.
> > 1. The reference-based evaluation metrics only measure the distance between the reconstructed image and reference images and none of them can reflect the degree of flicker-bandng. We truly need an additional evaluation experiments.
> > 2. We provide the results of user study in the following table, and our method gains the highest score. The score represents the percentage of rounds in which the model received preference out of the total number of evaluation rounds.
> >
> >     | Methods | RIFLE | PiSA-SR | MAT | InvSR | Step1X | ESDNet | NeRD-Rain | Notch Filter |
> >     |:--------:|:--------:|:--------:|:--------:|:--------:|:--------:|:--------:|:--------:|:--------:|
> >     | Score (%) |  82.4 | 14.7 | 2.7 | 0.02 | 0 | 0.1 | 0.08 |  0 |
> > 3. Searching for a metrics tailored for flicker-banding is also one of our future research directions.
>
> Q1-4: Luminance‑only Assumption Explanation.
> > A1-4: Thanks for your suggestion.
> >
> > 1. Similar to Q1-1, the dark straight FB covers the vast majority of scenarios and is our main research target. The color‑dependent artifacts are not a high-frequency scenario, and there is no need to simulate them.
> > 2. As for the domain‑gap analysis between simulated and real dark straight FB, we mention it in section 4.1.1 and present the comparison between the simulated and real FB  in Fig.4 in the main paper.

---

> ### Author Response · Authors · 2025-11-26
> **Call for further discussion with Reviewer qfME**
>
> > Dear Reviewer qfME,
> >
> > We sincerely appreciate the time you dedicate to reviewing our paper and providing your valuable feedback. We have carefully considered your comments and provided detailed responses:
> > 1. We provide explanation on **the scale and diversity of our dataset**.
> > 2. We present additional experimental results of **stronger task‑proximity baselines**.
> > 3. We add a **user study** across all the baselines to compensate for the disadvantage of existing evaluation metrics.
> > 4. We further explain our **luminance‑only assumption**.
> >
> > As the author–reviewer discussion phase is nearing its end, we would like to confirm whether our responses address your concerns. If any questions or suggestions remain, please feel free to let us know. We would be grateful to hear them.
> >
> > Thank you again for your time and valuable feedback.
> >
> > Best regards,
> >
> > Authors

---

> ### Author Response · Authors · 2025-11-28
> **Second call for discussion with Reviewer qfME**
>
> > Dear Reviewer qfME,
> >
> > We sincerely appreciate the time and effort you dedicated to reviewing our work, as well as your insightful comments. We have referred to your suggestions, conducting additional experiments of **stronger task‑proximity baselines** and a **user study as for additional evaluation scope**.
> >
> > Due to the approaching deadline for rebuttal and reviewer gzRh responding that our response can address the concerns on **the dataset and simulation process**, we also hope to discuss further with you whether or not your concerns have been addressed. Please let us know if you still have any unclear parts of our work.
> >
> > Best,
> >
> > Authors

---

### Author Response · Authors · 2025-11-21
**Response to all reviewers and area chairs for a summary**

Dear reviewers and area chairs,

We sincerely thank all reviewers and area chairs for their valuable time, constructive feedback, and appreciation of our contributions.

Across the reviews, several positive points were highlighted:
1. **Reviewer XrFB** commends that our work tackles a timely and practical problem with a complete, well-structured pipeline.
2. **Reviewers XrFB** and **tUpL** agree that our real-world paired dataset and simulation framework are valuable and novel, filling a gap in existing benchmarks.
3. **All reviewers** acknowledge the importance of formally defining the flicker-banding problem and establishing the first dataset, simulation process, and baseline model for this previously unaddressed degradation.

We have responded to each reviewer individually and would like to summarize our responses as follows:
1. **Reviewer qfME:** We added task-proximity baselines, a user study, and clarified dataset diversity to demonstrate the necessity and effectiveness of treating flicker-banding as an independent degradation.
2. **Reviewer tUpL:** We clarified the fundamental physical difference between flicker-banding and moiré patterns, provided demoiré model comparisons, and demonstrated that FB represents a distinct degradation pattern not addressed by existing methods.
3. **Reviewer gzRh:** We reinforced the novelty of establishing the first complete FB framework, showed why FPE and ML jointly enable substantial visual improvements beyond PSNR, clarified dataset versions to remove ambiguity, and supplied runtime/complexity analyses and additional baselines.
4. **Reviewer XrFB:** We validated our luminance-only and straight-stripe assumptions, provided out-of-distribution tests, and emphasized that our pipeline provides the first comprehensive FB framework for future research.

We extend our gratitude again to all reviewers and area chairs!

Best regards,

Authors

---

> ### Author Response · Authors · 2025-11-21
> **The revision in the main paper and the supplementary material**
>
> The revisions we make in the main paper and the supplementary material are marked with a blue background and are listed as follows.
>
> Our revision in the main paper:
> 1. We add our selection of image sizes for the test dataset to the table headers of table 1 and 2 in the main paper. As mentioned in the section 4.1.2, the real-world flicker-banding dataset has full-sized and cropped version. We denote it clearly to avoid misunderstanding of the difference in quantitative results between Tables 1 and 2.
>
> Our revision in the supplementary material:
> 1. We added the sentence "It's worth noting... more common scene" to section A of the supplementary material. This paper is the first to study striped stroboscopic flicker, and our goal is to showcase the diverse types of striped stroboscopic flicker observed during dataset construction. These findings highlight various flicker morphologies under extreme shooting conditions. However, in real-world scenarios, dark, straight stripes are most common, aligning with the underlying causes. Therefore, for flicker removal, focusing on scenes with dark, straight stripes covers most industrial applications. Severe FB scenes (e.g., black stripes, special shapes, colorful stripes) are theoretically impossible to fully recover from a single image, and we acknowledge our method's limitations in these cases. Researching extreme FB recovery from a single frame is impractical. For severe flicker-banding under very short exposure, we are exploring multi-frame solutions, which is a future research direction.
> 2. We provide a clarification on the quantitative results, mainly explaining that numerical metrics show smaller differences between models with large performance disparities due to the saturation to FB. Visual comparison is more important than the quantitative results when comparing model performance.
> 3. We provide additional experimental results in the section C of the supplementary material. We add simple notch filter (NF) aligned with our flicker-banding parameter estimator (FPE), ESDNet and NeRD-Rain. NF doesn't achieve any substantial effect and instead damages the image quality. Suppressing flicker in certain frequency bands also means losing the original image information at the same frequency band location. The irregular shape of the real-world FB can lead to biases in our FPE, making it difficult for the filter to accurately cover the frequency band where flicker exists. ESDNet is an outstanding demoiréing method accepted by ECCV 2022, while NeRD-Rain is an outstanding derain method accepted by CVPR 2024. Both of them are finetuned with our simulated dataset. RIFLE gains great significance over them in quantitative and visual results.
>
>     |   Methods    | PSNR $\uparrow$ | SSIM $\uparrow$ | ms-SSIM $\uparrow$ | LPIPS $\downarrow$ | DISTS $\downarrow$ | FSIM $\uparrow$ | GMSD $\downarrow$ |
>     | :-: | :-: | :-: | :-: | :-: | :-: | :-: | :-: |
>     |     LQ      |      19.43      |     0.5636      |       0.6364       |       0.3374       |       0.2213       |     0.7907      |      0.2091       |
>     | Notch Filter |      17.86      |     0.4067      |       0.5527       |       0.5340       |       0.3101       |     0.6734      |      0.2172       |
>     |    ESDNet    |      20.49      |     0.6077      |       0.7472       |       0.2782       |       0.2075       |     0.8424      |      0.1587       |
>     |  NeRD-Rain   |      20.30      |     0.6066      |       0.7447       |       0.2830       |       0.1950       |     0.8462      |      0.1702       |
>     | RIFLE (ours) |      20.66      |     0.6220      |       0.8067       |       0.2456       |       0.1723       |     0.8711      |      0.1433       |
> 4. We add demoiré model performance comparison in the case of Flicker-banding in section D of the supplementary material. The performance of the demoiré model largely decrease when both of the FB and moiré patterns exist. It indicates the difference between FB and moiré patterns and removing them simultaneously is also a valuable issue to be solved.
> 5. We provide a user study in the section E of the supplementary material. Because none of the evaluation metrics can reflect the degree of flicker-banding precisely, a user study is essential to compare different models' performance. The results of the user study are presented in following table, and our method gains the highest score. The score represents the percentage of rounds in which the model received preference out of the total number of evaluation rounds.
>     |  Methods  | RIFLE | PiSA-SR |  MAT  | InvSR | Step1X | ESDNet | NeRD-Rain | Notch Filter |
>     | :-: | :-: | :-: | :-: | :-: | :-: | :-: | :-: | :-: |
>     | Score (%) | 82.4  |  14.7   |  2.7  | 0.02  |   0    |  0.1   |   0.08    |      0       |
> 6. Although straight dark flicker-banding (FB) is our main research topic, we also provide out-of-distribution test results on other severe FB scenes in the section F of the supplementary material.

---

### Author Response · Authors · 2025-12-01
**Rebuttal-Discussion Summary and Reviewer Feedback Status**

Dear Program Chairs, Senior Area Chairs, and Reviewers:

We sincerely thank all program chairs, senior area chairs, area chairs, and reviewers (R1-qfME, R2-tUpL, R3-gzRh, R4-XrFB) for the time and expertise dedicated to evaluating our work. To assist the final decision process, we provide a concise summary of the rebuttal discussion and the current status of reviewer feedback.

### Summary of Rebuttal-Discussion
> Overall, the majority of reviewers recognize the core contributions of this paper, including:
> 1. Establishing the **first** complete framework for the flicker-banding (FB) problem, including **dataset, simulation, problem formulation, and baseline model**.
> 2. Providing a **real-world paired dataset** and a **simulation pipeline**, both viewed as valuable and novel additions that fill a gap in existing benchmarks.
> 3. Demonstrating that FB is a distinct and practically important degradation, fundamentally different from moiré patterns and **inadequately addressed** by existing methods.
>
> During the discussion phase, we addressed all raised concerns, including:
> 1. Adding **task-proximity baselines**, **demoiré comparisons**, **out-of-distribution tests**, and **runtime/complexity analyses**, responding to reviewers' requests for stronger empirical evidence.
> 2. Clarifying the **physical distinction** between FB and moiré, the **diversity and scale** of the dataset, and the assumptions underlying **our luminance-only and straight-stripe modeling**.
> 3. Providing a **user study** and a clarification on the **saturation of quantitative metrics**, explaining why **visual comparison is more informative** for FB removal and why small numerical gains should not be interpreted as inefficiency.

### Current Status of Reviewer Feedback
> We conclude the current status of reviewer feedbacks as follows:
> 1. **Reviewer gzRh**: Reviewer gzRh provide positive assessments, **acknowledging our contribution on the real-world dataset and simulation process**. However, he is still concerned about the effectiveness of our proposed component owing to the quantitative results. We provide further explanation on **the small improvement of the quantitative results while great enhancement in visual comparison**. Since the reviewer did not engage in the discussion phase further, we think the initial scoremay not reflect the actual situation and should be reconsidered accordingly.
> 2. **Reviewer qfME, tUpL and XrFB**: We have provided **clear explanations and additional experiments addressing all concerns** raised by these reviewer, which we believe sufficiently resolve the issues. Since thees reviewer did not participate in the discussion phase, the initial score may not fully reflect the updated clarifications, and we hope it can be reconsidered accordingly.

Our paper presents the first comprehensive treatment of the flicker-banding problem and provides models, datasets, and analyses intended to establish a foundation for future research in this unexplored degradation. We sincerely hope that the final decision may take these observations into careful consideration.

We sincerely hope that the final decision may take these observations into careful consideration.

Thank you again for your time, constructive feedback, and support.

Best regards,

Authors

---

### Meta-Review · Area_Chair_jhbP · 2025-12-23

**Summary:**

Paper targets flicker-banding (FB) in screen photography; contributions include paired real dataset, simulation pipeline, and a diffusion-based remover (RIFLE) with Masked Loss (ML) + Flicker-banding Prior Estimator (FPE).

Reviewers broadly agree the problem is important and the dataset/simulation pipeline are valuable; two reviewers are borderline (4/4), one is marginal accept (6), one is reject (2).

Decision is driven mainly by method contribution concerns: the claimed technical novelty (FPE) shows negligible quantitative gains over ML-only, raising doubts that RIFLE is more than “baseline + weighted loss”; and the dataset/benchmark size + assumptions (luminance-only, mostly straight stripes) limit generality.

Rebuttal clarifies table inconsistencies, adds baselines/user study/runtime, but does not convincingly establish FPE’s effectiveness. Overall: Reject.

**Reviewer Concerns:**

Addressed / mitigated:

Clarified cropped vs full-size splits; updated table headers; provided full-size main results.

Added notch filter, destriping/deraining (e.g., NeRD-Rain), demoiré comparisons (e.g., ESDNet), plus additional diffusion/transformer baselines already in the paper.

Added a user study emphasizing perceived banding removal.

Provided runtime and parameter comparison (PiSA-SR vs RIFLE) and discussed deployment setting.

Expanded physical explanation and provided demoiré results under FB.


Still outstanding:

Reviewer gzRh’s core point remains—after clarifications, FPE contributes ~0.02 dB PSNR on full-size data, suggesting it may be within training variance; evidence does not convincingly support the claim that FPE is a meaningful technical advance beyond ML.

Multiple reviewers note RIFLE is largely an integration of known components; with FPE’s limited measurable impact, the method contribution feels weak, leaving the dataset/simulation as the dominant contribution.

Benchmark remains small and focused; assumptions and acknowledged failures on more complex/colored/curved banding reduce claims of a general FB solution.

**Reviewer Scores:**

qfME (original 4): Estimated: 4 → 5.

tUpL (original 4): Estimated: 4 → 5.

gzRh (original 2): Estimated: 2 → 2.

XrFB (original 6): Estimated: 6 → 6.

---

### Decision · Program_Chairs · 2026-01-26

Reject